# Design and Synthesis of Conformationally Flexible Scaffold as Bitopic Ligands for Potent D_3_-Selective Antagonists

**DOI:** 10.3390/ijms24010432

**Published:** 2022-12-27

**Authors:** Ho Young Kim, Ji Youn Lee, Chia-Ju Hsieh, Michelle Taylor, Robert R. Luedtke, Robert H. Mach

**Affiliations:** 1Vagelos Laboratories, Department of Radiology, University of Pennsylvania, 1012, 231 S. 34th Street, Philadelphia, PA 19104, USA; 2Department of Pharmacology and Neuroscience, University of North Texas Health Science Center, Fort Worth, TX 76107, USA

**Keywords:** D_3_ receptor antagonists, metoclopramide, bitopic ligand, β-arrestin recruitment assay, computational chemistry

## Abstract

Previous studies have confirmed that the binding of D_3_ receptor antagonists is competitively inhibited by endogenous dopamine despite excellent binding affinity for D_3_ receptors. This result urges the development of an alternative scaffold that is capable of competing with dopamine for binding to the D_3_ receptor. Herein, an SAR study was conducted on metoclopramide that incorporated a flexible scaffold for interaction with the secondary binding site of the D_3_ receptor. The alteration of benzamide substituents and secondary binding fragments with aryl carboxamides resulted in excellent D_3_ receptor affinities (*K*i = 0.8–13.2 nM) with subtype selectivity to the D_2_ receptor ranging from 22- to 180-fold. The β-arrestin recruitment assay revealed that **21c** with 4-(pyridine-4-yl)benzamide can compete well against dopamine with the highest potency (IC_50_ = 1.3 nM). Computational studies demonstrated that the high potency of **21c** and its analogs was the result of interactions with the secondary binding site of the D_3_ receptor. These compounds also displayed minimal effects for other GPCRs except moderate affinity for 5-HT_3_ receptors and TSPO. The results of this study revealed that a new class of selective D_3_ receptor antagonists should be useful in behavioral pharmacology studies and as lead compounds for PET radiotracer development.

## 1. Introduction

Targeting D_2_ and D_3_ receptors has been studied for the treatment of neuropsychiatric disorders such as schizophrenia, and substance use disorders and addiction [1,2,3,4]. However, preferential localization of D_3_ receptors in limbic regions of the human brain suggested that D_3_ receptors may be a suitable target for developing therapeutics for treating neuropsychiatric disorders [5,6]. Other studies have demonstrated that this receptor plays a role in mediating the motivational actions of psychostimulants such as cocaine and amphetamine, and D_3_ antagonists have shown great promise in blocking cocaine self-administration in rodents and nonhuman primates [7,8]. The recent observation in the treatment of opioid use disorder has accelerated the need for the clinical evaluation of drugs targeting D_3_ receptors [9,10,11].

The development of dopamine D_3_-selective ligands continues to be a challenging area of medicinal chemistry research due to the high sequence homology of D_2_ and D_3_ receptor within the transmembrane (TM) domains (~79%) [12]. For developing receptor subtype selectivity, a “bitopic ligand” design has proven to be effective in the development of D_3_-selective compounds [13,14]. In this approach, a protonated basic amine in different scaffolds forms a salt bridge with Asp110^3.32^ of the D_3_ receptor in the orthosteric binding site (OBS), which is important for high binding affinity and the potency [15]. A secondary pharmacophore having an aromatic ring and appropriate linker group can result in high selectivity for the D_3_ receptor by the interaction with the secondary binding site (SBS) [16,17,18].

The first D_3_-selective scaffold contained an *N*-aryl piperazine moiety as the orthosteric binding fragment and an aryl carboxamide moiety with an alkyl linker as a secondary binding fragment [16,19,20,21,22,23,24,25]. This scaffold exhibited a sub nM binding affinity and good subtype selectivity for D_3_ receptors versus D_2_ receptors. However, these ligands also exhibited high binding affinity for other GPCRs (e.g., 5-HT or adrenergic receptors), which may lead the unwanted side effects [26,27,28]. Moreover, the in vivo properties of radiolabeled versions of this scaffold were not useful as PET radiotracers since they could not compete with endogenous dopamine for binding to the D_3_ receptor in vivo [21,29]. Since the replacement of substituents on benzamide or secondary binding fragments did not result in a significant change in properties of the *N*-aryl piperazine congeners, many groups have pursued other scaffolds, including azabicyclo [3.1.0]hexane [30,31,32], azaspiro alkane [33], diazaspiro alkane [34], tranylcypromine [35], or phenylcyclopropylmethylamine (PCPMA) [36]). However, these can be limited for clinical use under certain circumstances due to the poor bioavailability or toxicity [37,38,39] or are still under investigation. Recently, D_2_/D_3_ receptor agonist- and antagonist-modified bitopic ligands were developed based on (+)-PD128,907 or PF-592379 for selective agonist [40] and eticlopride for D_2_/D_3_ receptor ligands [41]. These compounds had comparatively low selectivity for D_3_ versus D_2_ receptors.

In the current study, we designed a new class of D_3_ receptor antagonist having the conformationally flexible scaffold of metoclopramide and the eticlopride-based benzamides (e.g., [^18^F]fallypride, *K*i D_2_ = 0.02 nM and D_3_ = 0.19 nM [42,43], IC_50_ = 1.7 nM [29]; [^11^C]FLB457, *K*i = 0.02 nM for D_2_/D_3_ receptors [44]) as lead compounds. Metoclopramide is largely used as an antiemetic; however, this compound also exhibited the low affinity for mixed D_2_/D_3_ receptors with the orthosteric binding fragment [45]. A combination of this flexible scaffold with well-established primary pharmocophore of the eticlopride-based benzamides was expected to achieve the high binding affinity and the potency for D_3_ receptors. Since the basic amine in this scaffold is structurally flexible without ring strain, the secondary binding fragment can be extended to strongly interact with the SBS while the orhosteric binding fragment remains bound to the OBS. Comprehensive screening was investigated for off-target interactions with other GPCRs; computational studies were also performed to provide the rational for the excellent potency of developed D_3_ receptor antagonists.

## 2. Results

### 2.1. Chemistry

Synthesis of 3-fluoropropyl or bromo analogs which have a dimethyl *tert*-amine with different length of carbon linker is shown in Figure 1. 5-(3-Fluoropropyl)-2,3-dimethoxybenzoic acid (**1a**) or 5-bromo-2,3-dimethoxybenzoic acid (**1b**) was conjugated with secondary amine Boc-protected *tert*-butyl (2-aminoethyl)(propyl)carbamate by amide coupling in a quantitative yield. After the removal of the Boc protecting group, free amine **3a** or **3b** was *N*-alkylated with 2-(3-bromopropyl) or 2-(4-bromobutyl)-1,3-dioxolane. Dioxolanes **4a** to **4d** were hydrolyzed using aqueous 4 N HCl at RT to give the aldehyde **5a** to **5d** which were conjugated with dimethylamine via reductive amination. 5-(3-Fluoropropyl) or 5-bromo-2,3-dimethoxybenzamide analogs having the dimethylamine moiety (**6a**–**d**) were obtained in 37–47% yield over the two-step synthesis.

The next series focused on preparing analogs having a spacer group with an aromatic ring system for interacting with the SBS. For the aromatic ring moiety, we tested 4-(thiophen-2-yl)benzamide or 4-methyl-5-phenyl-4*H*-1,2,4-triazole-3-thiol. The 4-(Thiophen-2-yl)benzamide fragment was chosen from our previous results. This aromatic ring system was observed in LS-3-134 and other structural congeners having a high D_3_ affinity and excellent selectivity versus the D_2_ receptor [46,47,48]. **3b** was *N*-alkylated with *N*-(3-bromopropyl) or *N*-(4-bromobutyl)phthalimide and then the protecting phthalimide **7a** or **7b** was hydrolyzed using hydrazine hydrate by heating for 3 h to give primary amine **8a** or **8b** (Figure 2). 4-(Thiophen-2-yl)benzoic acid was converted to the corresponding acyl chloride using thionyl chloride at RT followed by treatment with **8a** or **8b** to give **9a** or **9b** in 50% or 20% yield, respectively. The triazole-thiol ether analogs were prepared by reduction of **5c** and **5d** to give alcohols **10a** and **10b**, which were converted to **11a** and **11b** using Mitsunobu reaction. The desired products **11a** or **11b** were obtained in 16% and 28% yield, respectively (Figure 2).

Inspection of the structure of **9a** reveals that two different benzamide fragments which share the *tert*-amine are capable of interacting with the OBS of the D_2_ and D_3_ receptors. Therefore, fragments **12** and **14** were synthesized for evaluation in in vitro binding studies (Figure 3). **12** was synthesized by *N*-methylation from the secondary amine **3b** in 19% yield. For **14**, 4-(thiophen-2-yl)benzoic acid was conjugated with 3-bromopropylamine through acyl chlorination followed by *N*-alkylated with *N*-methylethanamine.

The next series probed the size of substituents on the *tert*-amine group. The pendent synthons for allyl (**15a**) and 4-fluorobenzyl (**15b**) were prepared from ethylenediamine (Figure 4). For the synthesis of **15a**, one of the primary amines was protected with a trifluoroacetyl group and the other primary amine alkylated with allyl bromide. The secondary amine was protected as a *N*-Boc and the trifluoroacetyl group was removed. **15b** was synthesized in a similar method with **15a** except a reductive amination with 4-fluorobenzaldehyde was used. The prepared synthon **15a** or **15b** was conjugated with **1b**, and the *N*-Boc was removed to give intermediates **17a**,**b**. These intermediates were treated with *N*-propylphthalimide to give **18a**,**b**. Removal of the phthalimide group with hydrazine hydrate gave corresponding *N*-propyl intermediate **19a** (via reduction of the *N*-allyl group) and the 4-fluorobenzyl analog **19b**. The intermediates **19a** and **19b** were conjugated with 4-(thiophen-2-yl)benzoic acid to give the desired products **20a** and **20c** in 71% or 32% yield, respectively. For the *N*-allyl analog, **17a** was directly *N*-alkylated with **13** to give **20b** in 21% yield.

To investigate the nature of the aromatic moiety for binding to the SBS, aryl carboxamides **21b**–**j** were synthesized using the same method described for the synthesis of **9a** but using different aryl carboxylic acids and the naphthamide **21a** was synthesized using 2-naphthoyl chloride using in the basic condition (Figure 5). The desired benzamide analogs were obtained in yields ranging from 20 to 86%, respectively. The purity of all investigated compounds was confirmed prior to analysis and was greater than 95% on a 2695 Alliance LC-MS (Appendix A).

### 2.2. SAR Study

Two different assays were used to evaluate the properties of the analogs described above. The receptor binding affinity was measured by radioligand binding assays using [^125^I]IABN with D_2_ or D_3_ receptors highly expressed HEK293 cells [49]. The functional activity of the analogs was determined using a β-arrestin recruitment assay. The assay was initially conducted in agonist binding mode to confirm that they function as antagonists at the D_3_ receptor. Once this efficacy was confirmed, the assay was conducted in antagonist mode to determine the ability of the antagonist to compete with dopamine at the D_3_ receptor. The results of the antagonist mode assay are reported as IC_50_ values [50,51,52]. Imax values were individually calculated from the assay and reliable with over 50% inhibition.

The first series of compounds evaluated were those synthesized in Figure 1 and Figure 2 (Table 1). The dimethyl amino analogs **6a**–**d** displayed a relatively low binding affinity for both D_2_ and D_3_ receptors. These data suggest that a basic amine moiety in the spacer group reduces affinity at both receptors. The observation that **6d** had a 10-fold higher affinity than its structural congener **6b** indicates that the Br-substituent is more preferred in the OBS than the corresponding fluoropropyl substituent. Compounds **9a**,**b** and **11a**,**b**, which have aromatic groups in the SBS, displayed a higher affinity at both D_2_ and D_3_ receptors. The 4-(thiophen-2-yl)benzamide analogs were more potent at the D_3_ receptor than the corresponding 4-methyl-5-phenyl-4*H*-1,2,4-triazole-3-thiol analogs. These data suggest that benzamides are preferred in the SBS of the D_3_ receptor for this scaffold. It is of interest to note that **9a** had ~170-fold higher affinity at the D_3_ versus the D_2_ receptor.

It is important to note that **9a** has two different modes in which it can bind to the D_3_ receptor. The first mode has the bromobenzamide moiety binding to the OBS and the 4-(thiophen-2-yl)benzamide binding to the SBS. The second mode has the 4-(thiophen-2-yl)benzamide binding to the OBS and the bromobenzamide moiety binding to the SBS. In vitro binding studies revealed that fragment **12** showed non-selectively high *K*i values at both of dopamine receptor subtypes (*K*i D_2_ = 89.2 ± 5.6 nM, D_3_ = 21.8 ± 5.1 nM), whereas **14** did not show any binding affinity at D_2_ and D_3_ receptors (*K*i D_2_ > 1000 nM and D_3_ > 1000 nM). Moreover, the β-arrestin recruitment assay indicated that compound **18** is very potent for the D_3_ receptor (IC_50_ = 4.6 ± 1.2 nM). These data are consistent with the first mode that the bromobenzamide moiety binds to the OBS and the 4-(thiophen-2-yl)benzamide binds to the SBS.

Table 2 shows the effect of the size of the *N*-alkyl group in the *tert*-amine on the D_2_ and D_3_ receptor binding. Our results indicate that the *N*-ethyl substituent **9a** showed the highest binding affinity and subtype selectivity at the D_3_ receptor versus the D_2_ receptor. There was a slight decrease in affinity in going from propyl to allyl groups, whereas the 4-fluorobenzyl group resulted in a large loss in affinity at both D_2_ and D_3_ receptors (Table 2). When the size of substituents was increased, the binding affinity and subtype selectivity was decreased. This reduction in affinity also translated to the β-arrestin recruitment assay. That is, there was a trend of decreased potency in the order of **9a** (IC_50_ = 14.0 ± 7.4 nM) > **20a** (IC_50_ = 26.5 ± 12.9 nM) > **20b** (IC_50_ = 51.6 ± 40.8 nM). Based on this SAR study, the *N*-ethyl group is the preferred alkyl group with respect to binding to the OBS.

A number of compounds were prepared to explore the nature of the interaction between the aromatic ring and the SBS. Previous studies with the *N*-aryl piperazine analogs revealed that a wide range of aromatic rings are tolerated in the SBS with respect to D_3_ affinity, but the overall D_3_ versus D_2_ selectivity can be influenced by the nature of this interaction. The results of this study are shown in Table 3. All compounds had good affinity at D_3_ receptors, with *K*i values ranging between 0.8 and 13.2 nM. The D_2_ affinities ranged between 107 and 525 nM, resulting in a D_3_ selectivity ratio (i.e., D_2_/D_3_ ratio) ranging from 22.1- to 180-fold. The effect of the nature of the aromatic ring in the SBS on the ability of the antagonist to compete with dopamine in the β-arrestin assay was somewhat unexpected. For example, both **21a** and **21c** have ~1 nM affinity for the D_3_ receptor in the radioligand binding assay, but the potency of **21c** in the β-arrestin recruitment assay was 10-fold higher than that of **21a** (IC_50_ = 1.3 vs. 16.4 nM).

### 2.3. Molecular Docking and Molecular Dynamics Simulations (MDS)

To understand the favorable binding profiles of the metoclopramide analogs, molecular docking and MDS studies were performed using different *N*-alkyl compounds (**9a**, **20a**, **20b**, **20c** and **21c**) with the D_3_ receptor (PDB: 3PBL) (Table 4). These compounds were chosen because they are close structural analogs and have a wide range in D_3_ receptor affinity (1–300 nM). As reported in previous studies [13,29,53], the binding pose that formed a bridge hydrogen bond between the carboxylate of ASP110^3.32^ and the protonated nitrogen was considered to be critical for high binding affinity for the D_3_ receptor. The distance between the protonated nitrogen ranged between 2.6 and 2.9 Å, and **9a** was found to have the closest interaction (2.6 Å). The estimated binding energies were not significantly different for each compound (−9.74 to −10.22 kcal/mol). Therefore, the difference in D_3_ affinity of the five compounds cannot be explained by the distance between ASP110^3.32^ and the protonated nitrogen atom, and the calculated binding energies from docking studies.

In MDS studies, the root mean square distance (RMSD) was calculated over 50–200 ns in five copies of the MDS production (Table 4). The first time frame (0 ns) of the production run was used as the reference position to determine the stability of each compound in the binding site. **21c** presented the lowest standard deviation of RMSD (2.45 ± 0.49 Å) indicating the least amount of movement in the binding site. A relatively higher amount of motion (3.00 ± 0.82 Å) with **20c** is consistent with the lower binding affinity for D_3_ receptors. These results indicate the MDS studies correlate better with D_3_ affinity than the results of docking studies.

The representative binding pose of the MDS production run is displayed in Figure 1. Within the OBS of the D_3_ receptor, all the selected compounds were engaged in multiple interactions. The hydrogen bond with ASP110^3.32^ and π staking interactions with PHE345^6.52^ were observed with all five compounds. However, a halogen bond between VAL189^5.39^ and the bromine of the 5-bromo-2,3-dimethoxybenzene moiety was observed for **9a**, **21c**, and **20c** (Figure 1a,b,e, respectively). It is of interest to note the **21c**, the most potent compound in the β-arrestin recruitment assay, which predicts the ability to compete with endogenous dopamine, had a cation–π interaction between the protonated nitrogen and PHE106^3.28^ residue (Figure 1b).

The summary of overall frequency of contacts from the MDS studies, including hydrophobic interactions, hydrogen bonds, the salt bridge, halogen bonds, and π-interactions, is shown in Figure 2. All five compounds formed stable interactions (frequency of contact > 0.6) with most of residues in the OBS (i.e., ASP110^3.32^, VAL111^3.33^, CYS114^3.36^, SER196^5.46^, PHE345^6.51^, and THR369^7.39^). The frequency of all interactions in the OBS of **20c**, which exhibited the lowest binding affinity for D_3_ receptors, was lower than the higher-affinity compounds. As mentioned above, **21c** showed a high frequency of contacts with PHE106^3.28^ including approximately 95% of hydrophobic interactions and 10% of cation–π interactions over the MDS production runs.

Consistent with previous modeling studies, the formation of key interactions between ASP110^3.32^ and the protonated nitrogen of the ligand stabilized the binding pose of **9a**, **20a**, **20b**, and **20c** (frequency of contact > 0.998) by 97.8% to 99.4% of the hydrogen bond formation. However, the frequency of contacts between ASP110^3.32^ and **20c** was relatively lower (frequency of contact = 0.990) and formed only 68.6% of hydrogen bonds over the MDS production runs.

In the SBS, **9a**, **20a** and **21c** that exhibited high subtype selectivity, presented a moderate to high probability (frequency of contacts = 0.4–0.9) of interaction with VAL86^2.61^, LEU89^2.64^, GLY93^EL1^, and GLY94^EL1^. In addition, the pyridine of **21c** formed a hydrophobic interaction with GLU90^2.65^ (frequency of interaction = 0.563). In contrast to our expectations, 90% of the hydrophobic interactions that formed with VAL86^2.61^ were from the 4-fluorobenzyl group whereas 10% of the interactions were from the 4-(thiophen-2-yl)benzamide moiety.

The average frequency of the overall interactions in the binding sites (i.e., OBS and SBS) was correlated with D_3_ receptor binding affinity (r = −0.8756, and *p* = 0.0517). In addition, the average frequency of interaction in the OBS was significantly correlated with the IC_50_ values from the β-arrestin recruitment assay (r = −0.9934, and *p* = 0.0066).

### 2.4. Comprehensive Screening for Other GPCRs

Based on the results in the dopamine receptor radioligand binding assays, nine flexible-based compounds were selected for further evaluation for off-target binding with other GPCRs through the Psychoactive Drug Screening Program (PDSP) (Appendix A) [54]. Previous studies with the *N*-aryl piperazine analogs showed high binding affinity for serotonin 5-HT_1A_ and 5-HT_2B_ receptors. For example, many of the *N*-aryl piperazine-based analogs that our group developed in the past for either the D_2_ or D_3_ receptor had high affinity for the 5-HT_1A_ receptor [55,56,57,58]. It is of interest to note that none of the panel submitted for evaluation had a high affinity for the 5-HT_1A_ receptor or any of the other GPCRs in the screening assay (Appendix A). Compounds **21a**, **21c**, and **21i** had modest affinity for the 5-HT_3_ receptor (*K*i values 29–58 nM). Furthermore, a relatively high affinity of compounds **20a**, **21a**, **21e**, **21g**, and **21i** for the peripheral benzodiazepine receptor (PBR) was observed. This mitochondrial-based protein is typically used as a target for imaging neuroinflammation. The results of the PDSP-binding assays also confirmed the data obtained in our lab for the binding of this panel of nine compounds to D_2_ and D_3_ receptors (Appendix A).

## 3. Discussion

The goal of the current study was to identify a new scaffold for D_3_-selective antagonists that must display a high affinity and selectivity for D_3_ versus D_2_ receptors in the radioligand binding assays, but also a high potency in a β-arrestin recruitment assay, which measures the ability of a compound to compete with dopamine in binding to the D_3_ receptor [21,29]. Previous studies have shown that a PET radiotracer developed in our lab having a high affinity for the D_3_ receptor (Kd~50 pM) and excellent selectivity versus the D_2_ receptor (>150-fold) was not able to image D_3_ receptors in vivo without pretreatment with drugs that reduce synaptic levels of dopamine [59].

For the current study, we chose metoclopramide as the lead compound for our SAR studies. Metoclopramide was chosen as the lead compound for this study because it has a modest affinity for both D_2_ and D_3_ receptors and it should be possible to make analogs of this compound having an improved D_3_ binding affinity while minimizing D_2_ receptor affinity by interacting with the SBS. The results of SAR indicated that 5-bromo-2,3-dimethoxybenzamide, the moiety from FLB457, was more favorable for binding to the OBS, which is important for determining affinity for both D_3_ and D_2_ receptors. The size of fragments in **9a**,**b** or **11a**,**b** that interact with the SBP residues of the D_3_ receptor are important for high selectivity for D_3_ versus D_2_ selectivity [60]. It is of interest to note that the appropriate length of linker between the basic amine and the secondary binding fragment was one carbon shorter than other known D_3_ receptor antagonists such as *N*-arylpiperazine congeners. The D_3_ receptor binding affinity was also affected by steric hindrance of the substituent on the basic amine.

A number of the compounds reported here exhibited excellent D_3_ binding affinity (ranging from 0.8 to 13.2 nM) and excellent selectivity (22.1- to 180-fold) for D_3_ vs. D_2_ receptors. Although analogs such as **9a**, **21a**, **21d**, and **21g** exhibit high binding affinity and subtype selectivity for the D_3_ receptor, **21c** was identified as the best-in-series candidate because of its high D_3_ affinity and selectivity, and excellent potency in the β-arrestin recruitment assay (IC_50_ = 1.3 nM). This IC_50_ value was comparable with fallypride that is widely used as a non-selective PET probe for D_2_/D_3_ receptors and can bind to D_3_ receptors in the presence of endogenous dopamine (fallypride, IC_50_ = 1.7 nM) [29]. Moreover, the computational modeling studies demonstrated that the high potency of **21c** may result from the short distance of the bridge-bond with ASP110^3.32^ and the high-frequency contacts between **21c** and residues in the OBS and SBS in the D_3_ receptor.

Since metoclopramide was previously used in drug development and led to the identification of compounds having a diverse range of pharmacologic activity including mixed 5-HT_3_ antagonists/5-HT_4_ agonists (e.g., zacopride, BRL 24682) and D_2_ antagonists (e.g., clebopride, BRL 25594) [45], there was a concern that the conformational flexibility of our compounds could result in significant off-target bindings to other G-proteins. By the comprehensive screening from PDSP, these compounds possess minimal affinity for other GPCRs except a moderate affinity for 5-HT_3_ receptors (29–58 nM). Interestingly, **21d**, which has an indole carboxamide as a secondary binding fragment, exhibited nM binding affinity for the histamine H_1_ receptor (0.95 nM). Other compounds acquired affinity for the translocator protein (TSPO); however, it is not clear if this off-target binding would be problematic for using these compounds in D_3_ receptor binding assays or behavioral studies. Further studies are ongoing in our lab to prepare radiolabeled versions of **21c** for imaging D_3_ receptors in the brain, and SAR studies are being conducted that aim to improve the properties of this new scaffold as a means of identifying potential D_3_ receptor selective PET radiotracers.

## 4. Materials and Methods

### 4.1. General

5-(3-Fluoropropyl)-2,3-dimethoxybenzoic acid (**1a**) was prepared from methyl 5-allyl-3-methoxy salicylate via methylation for phenol, oxidation of the allyl group, fluorination, and hydrolysis of methyl ester [61]. 5-Bromo-2,3-dimethoxybenzoic acid (**1b**) was prepared from 5-bromo-2-hydroxy-3-methoxy benzoic acid via methylation for phenol and oxidation of aldehyde to carboxylic acid using silver catalyst [62]. For the OBS binding, ***tert***-butyl (2-aminoethyl) ethylcarbamate was prepared from ***N***-ethylethylenediamine via the primary amine protection, the secondary amine protection and the primary amine de-protection according to the literature [63]. 2-(2-Bromoethyl)-1,3-dioxolane and 2-(3-bromopropyl)-1,3-dioxolane were prepared via reduction followed by cyclization [64]. The other reagents and solvents were purchased from Sigma-Aldrich, TCI, Matrix Scientific, Advanced chemtech, Fisher chemical, Ambeed, Chembridge corporation, Acros organics, and Decon laboratories and used as received (Appendix A). Reactions were monitored by thin layer chromatography (TLC) using TLC silica gel 60W F254S plates and the spots were detected under UV light (254 nm) or developed using ninhydrin. Flash column chromatography was carried out on a Biotage Isolera One with a dual wavelength UV-vis detector. ^1^H and ^13^C NMR spectra were obtained on a Bruker NEO-400 spectrometer (Bruker, Billerica, MA, USA). Chemical shifts (δ) were recorded in parts per million (ppm) relative to the deuterated solvent as an internal reference. Mass spectra (m/z) were recorded on a 2695 Alliance LC-MS (Waters Corporation, Milford, MA, USA) using positive electrospray ionization (ESI**^+^**). High resolution mass spectra (HRMS, *m*/*z*) were acquired on a waters LCT premier mass spectrometer (Waters Corporation, Milford, MA, USA). PathHunter^TM^ β-arrestin recruitment assay kit and the Chinese hamster ovary CHO-K1 cell line were purchased from DiscoverX (Fremont, CA, USA).

### 4.2. Chemistry

*tert*-Butyl ethyl(2-(5-(3-fluoropropyl)-2,3-dimethoxybenzamido)ethyl)carbamate (**2a**) In a mixture of **1a** (1 g, 4.13 mmol), ***tert***-butyl (2-aminoethyl) ethylcarbamate, (1.55 g, 8.26 mmol) and HBTU (1.55 g, 6.2 mmol) in DMF (20 mL), DIPEA (1.08 mL, 6.2 mmol) was added. The mixture was stirred for 24 h at RT. After completion of the reaction, the mixture was diluted with EtOAc and washed with water and brine. The volatiles were removed under reduced pressure and the crude product was purified by flash chromatography on silica gel (EtOAc/hexane = 1:2) to afford **2a** (1.15 g, 68% yield) as a yellow oil. (^1^H NMR, 400 MHz, CDCl**_3_**): δ = 8.12 (br, 1H), 7.45 (s, 1H), 6.81 (s, 1H), 4.43 (t, *J* = 5.9 Hz, 1H), 4.31 (t, *J* = 5.9 Hz, 1H), 3.81 (s, 3H), 3.80 (s, 3H), 3.53 (q, *J* = 6.1 Hz, 2H), 3.37 (t, *J* = 6.3 Hz, 2H), 3.22 (d, *J* = 6.2 Hz, 2H), 2.66 (t, *J* = 7.4 Hz, 2H), 2.00–1.87 (m, 2H), 1.37 (s, 9H), 1.05 (t, *J* = 7.0 Hz, 3H); (^13^C NMR, 100 MHz, CDCl**_3_**): δ = 165.4, 152.3, 145.8, 137.2, 83.6, 82.0, 79.4, 61.1, 56.0, 45.7, 31.8, 31.6, 31.0**_3_**, 30.9**_8_**, 28.2; ESI-MS *m/z* calculated for C_21_H_34_FN_2_O_5_^+^ [M+H]**^+^** 413.5; found 413.6.

*tert-*Butyl (2-(5-bromo-2,3-dimethoxybenzamido)ethyl)(ethyl)carbamate (**2b**) **2b** was synthesized using **1b** (5 g, 19.15 mmol) in the same procedure as **2a** and purified by flash chromatography on silica gel (EtOAc/hexane = 1:2) to afford **2b** (8 g, 97% yield) as a yellow oil. (^1^H NMR, 400 MHz, CD_3_CN): δ = 7.96 (br, 1H), 7.57 (d, *J* = 2.2 Hz, 1H), 7.24 (d, *J* = 2.3 Hz, 1H), 3.85 (s, 3H), 3.83 (s, 3H), 3.48 (q, *J* = 6.0 Hz, 2H), 3.37 (t, *J* = 6.0 Hz, 2H), 3.23 (q, *J* = 7.0 Hz, 2H), 1.38 (s, 9H), 1.07 (t, *J* = 7.0 Hz, 3H) (^13^C NMR, 100 MHz, CD_3_CN): δ = 164.8, 154.9, 147.9, 125.1, 119.2, 117.1, 79.8, 62.0, 57.2, 39.4, 28.6; ESI-MS *m/z* calculated for C_18_H_27_BrN_2_O_5_^+^ [M]^+^ 431.3; found 431.4.

*N*-(2-(Ethylamino)ethyl)-5-(3-fluoropropyl)-2,3-dimethoxybenzamide (**3a**) In a solution of **2a** (1.15 g, 2.79 mmol) in 15 mL of CH**_2_**Cl**_2_**, TFA (15 mL, 196 mmol) was slowly added at 0 °C. The reaction mixture was warmed to RT and stirred for 1 h. The volatiles were removed followed by the residue was dissolved in CH_2_Cl_2_ and the organic layer washed by aq saturated NaHCO_3_ solution. The inorganic layer was extracted by CH_2_Cl_2_ and the combined layer was washed by brine, dried over anhydrous MgSO_4_, filtered and concentrated in vacuo to afford **3a** (850 mg, 98% yield) as a yellow oil. The crude product was used for the next step without further purification. (^1^H NMR, 400 MHz, CD**_3_**CN): δ = 8.26 (br, 1H), 7.36 (d, *J* = 2.2 Hz, 1H), 7.02 (d, *J* = 2.1 Hz, 1H), 4.52 (t, *J* = 6.0 Hz, 1H), 4.40 (t, *J* = 6.0 Hz, 1H), 3.86 (s, 3H), 3.84 (s, 3H), 3.43 (q, *J* = 5.8 Hz, 2H), 2.78 (t, *J* = 6.1 Hz, 2H), 2.71 (t, *J* = 7.6 Hz, 2H), 2.65 (q, *J* = 7.1 Hz, 2H), 2.00–1.96 (m, 2H), 1.37 (s, 9H), 1.08 (t, *J* = 7.1 Hz, 3H); (^13^C NMR, 100 MHz, CD**_3_**CN): δ = 165.9, 153.9, 146.8, 138.7, 128.1, 122.4, 116.7, 85.2, 61.8, 56.8, 49.4, 44.5, 40.2, 32.9, 31.8, 15.7; ESI-MS *m/z* calculated for C_16_H_26_FN_2_O_3_^+^ [M+H]**^+^** 313.4; found 313.5.

Bromo-*N*-(2-(ethylamino)ethyl)-2,3-dimethoxybenzamide (**3b**) **3b** was synthesized using **2b** (8 g, 18.55 mmol) in the same procedure as **3a** and purified by flash chromatography on silica gel (CH_2_Cl_2_/7 N NH_3_ in MeOH = 20:1) to afford 3b (6 g, 98% yield) as a yellow oil. (^1^H NMR, 400 MHz, CD_3_CN): δ = 8.24 (br, 1H), 7.60 (d, *J* = 2.4 Hz, 1H), 7.26 (d, *J* = 2.4 Hz, 1H), 3.86 (s, 3H), 3.85 (s, 3H), 3.43 (q, *J* = 5.7 Hz, 2H), 2.80 (t, *J* = 6.0 Hz, 2H), 2.66 (q, *J* = 7.1 Hz, 2H), 1.08 (t, *J* = 7.1 Hz, 3H) (^13^C NMR, 100 MHz, CD_3_CN): δ = 164.5, 155.0, 148.0, 129.8, 125.2, 119.3, 117.2, 62.0, 57.2, 49.1, 44.4, 40.1, 15.4; ESI-MS *m/z* calculated for C_13_H_21_BrN_2_O_3_^+^ [M]^+^ 331.2; found 331.4.

*N*-(2-((3-(1,3-Dioxolan-2-yl)propyl)(ethyl)amino)ethyl)-5-(3-fluoropropyl)-2,3-dimethoxybenzamide (**4a**) In a solution of **3a** (300 mg, 0.96 mmol) in 9.6 mL of MeCN, 2-(3-bromopropyl)-1,3-dioxolane (281 mg, 1.44 mmol) and Na**_2_**CO**_3_** (254 mg, 2.4 mmol) were added. The reaction mixture was stirred for 24 h at 65 °C and another 1 eq of 2-(3-bromopropyl)-1,3-dioxolane (187 mg, 0.96 mmol) was added. The reaction mixture was stirred for 48 h at 65 °C. After the completion of the reaction which was checked by TLC, aq saturated NaHCO_3_ solution was added and the crude product was extracted with EtOAc. The organic layer was washed by brine, dried over MgSO_4_, filtered and concentrated in vacuo. The crude product was purified by flash chromatography on silica gel (CH_2_Cl_2_/MeOH = 10:1) to afford 10a (240 mg, 59% yield) as a yellow oil. (^1^H NMR, 400 MHz, CD_3_CN): δ = 8.19 (br, 1H), 7.30 (d, *J* = 2.1 Hz, 1H), 6.94 (d, *J* = 2.0 Hz, 1H), 4.70 (t, *J* = 4.4 Hz, 1H), 4.44 (t, *J* = 6.0 Hz, 1H), 4.32 (t, *J* = 6.0 Hz, 1H), 3.79–3.75 (m, 8H), 3.69–3.65 (m, 2H), 3.35 (q, *J* = 5.8 Hz, 2H), 2.63 (t, *J* = 7.6 Hz, 2H), 2.55 (t, *J* = 6.1 Hz, 2H), 2.51 (q, *J* = 7.1 Hz, 2H), 2.44 (t, *J* = 7.2 Hz, 2H), 1.89–1.86 (m, 2H), 1.56**–**1.42 (m, 4H), 0.94 (t, *J* = 7.1 Hz, 3H) (^13^C NMR, 100 MHz, CD_3_CN): δ = 165.2, 153.4, 146.4, 138.2, 127.4, 122.0, 116.2, 104.7, 84.7, 83.1, 65.1, 61.4, 56.3, 53.3, 52.6, 47.6, 37.8, 32.4, 32.2, 32.0, 31.4, 31.3, 22.1, 15.6; ESI-MS *m/z* calculated for C_22_H_36_FN_2_O_5_^+^ [M+H]^+^ 427.5; found 427.6.

*N*-(2-((4-(1,3-Dioxolan-2-yl)butyl)(ethyl)amino)ethyl)-5-(3-fluoropropyl)-2,3-dimethoxybenzamide (**4b**) **4b** was synthesized using **3a** (346 mg, 1.10 mmol) and 2-(4-bromobutyl)-1,3-dioxolane (577 mg, 2.76 mmol) in the same procedures as 4a and obtained 282 mg (58% yield) as a yellow oil. (^1^H NMR, 400 MHz, CD_3_CN): δ = 8.34 (br, 1H), 7.41 (d, *J* = 2.0 Hz, 1H), 7.06 (d, *J* = 2.0 Hz, 1H), 4.75 (t, *J* = 4.4 Hz, 1H), 4.56 (t, *J* = 6.0 Hz, 1H), 4.44 (t, *J* = 6.0 Hz, 1H), 3.90–3.88 (m, 8H), 3.79–3.76 (m, 2H), 3.49 (q, *J* = 5.7 Hz, 2H), 2.77–2.66 (m, 6H), 2.57 (t, *J* = 6.5 Hz, 2H), 2.10–2.00 (m, 2H), 1.64–1.52 (m, 4H), 1.46–1.38 (m, 2H), 1.09 (t, *J* = 7.1 Hz, 3H) (^13^C NMR, 100 MHz, CD_3_CN): δ = 165.3, 153.4, 146.4, 138.2, 127.4, 121.9, 116.2, 104.7, 84.7, 83.1, 65.0, 61.4, 56.3, 53.4, 52.6, 47.8, 37.6, 34.0, 32.4, 32.2, 31.4, 31.3, 22.2; ESI-MS *m/z* calculated for C_23_H_38_FN_2_O_5_^+^ [M+H]^+^ 441.6; found 441.7.

*N*-(2-((3-(1,3-Dioxolan-2-yl)propyl)(ethyl)amino)ethyl)-5-bromo-2,3-dimethoxybenzamide (**4c**) **4c** was synthesized using **3b** (320 mg, 0.97 mmol) in the same procedures as 4a and obtained 269 mg (62% yield) as a yellow oil. (^1^H NMR, 400 MHz, CD_3_CN): δ = 8.23 (br, 1H), 7.62 (d, *J* = 2.4 Hz, 1H), 7.25 (d, *J* = 2.4 Hz, 1H), 4.77 (t, *J* = 4.8 Hz, 1H), 3.85–3.82 (m, 8H), 3.75–3.71 (m, 2H), 3.40 (q, *J* = 5.7 Hz, 2H), 2.59 (t, *J* = 6.1 Hz, 2H), 2.55 (q, *J* = 7.1 Hz, 2H), 2.48 (t, *J* = 7.2 Hz, 2H), 1.59–1.49 (m, 4H), 0.99 (t, *J* = 7.1 Hz, 3H) (^13^C NMR, 100 MHz, CD_3_CN): δ = 164.0, 154.9, 148.0, 129.6, 125.3, 119.3, 117.2, 105.2, 65.5, 62.0, 57.2, 53.7, 52.9, 48.0, 38.4, 32.5, 22.7, 12.2; ESI-MS *m/z* calculated for C_19_H_29_BrN_2_O_5_^+^ [M]^+^ 445.4; found 445.5.

*N*-(2-((4-(1,3-Dioxolan-2-yl)butyl)(ethyl)amino)ethyl)-5-bromo-2,3-dimethoxybenzamide (**4d**) **4d** was synthesized using **3b** (309 mg, 0.93 mmol) in the same procedures as 4a and obtained 412 mg (96% yield) as a yellow oil. (^1^H NMR, 400 MHz, CD_3_CN): δ = 8.25 (br, 1H), 7.62 (d, *J* = 2.4 Hz, 1H), 7.25 (d, *J* = 2.4 Hz, 1H), 4.70 (t, *J* = 4.8 Hz, 1H), 3.85–3.82 (m, 8H), 3.74–3.70 (m, 2H), 3.41 (q, *J* = 5.8 Hz, 2H), 2.61 (t, *J* = 6.1 Hz, 2H), 2.57 (q, *J* = 7.1 Hz, 2H), 2.47 (t, *J* = 7.1 Hz, 2H), 1.58–1.53 (m, 2H), 1.51–1.44 (m, 2H), 1.40–1.32 (m, 2H), 1.00 (t, *J* = 7.1 Hz, 3H) (^13^C NMR, 100 MHz, CD_3_CN): δ = 164.1, 154.9, 148.0, 129.6, 125.3, 119.3, 117.1, 105.2, 65.5, 62.0, 57.2, 53.8, 52.9, 48.1, 38.3, 34.6, 27.7, 22.8, 12.0; ESI-MS *m/z* calculated for C_20_H_31_BrN_2_O_5_^+^ [M]^+^ 459.4; found 459.5.

*N*-(2-(Ethyl(4-oxobutyl)amino)ethyl)-5-(3-fluoropropyl)-2,3-dimethoxybenzamide (**5a**) In a solution of **4a** (86 mg 0.2 mmol), in 2 mL of THF, 2 mL of aq 4 N HCl was slowly added. The reaction mixture was stirred for 3 h at RT and then, neutralized by 4 mL of aq 2 N NaOH solution. The crude product was extracted with EtOAc and the organic layer was washed by aq saturated NaHCO_3_ solution, water and brine. The organic layer was dried over MgSO_4_, filtered and concentrated in vacuo to afford 11a (74 mg, 96% yield) as a colorless oil. 11a was used without further purification for the next step. ESI-MS *m/z* calculated for C_20_H_32_FN_2_O_4_^+^ [M+H]^+^ 383.5; found 383.5.

*N*-(2-(Ethyl(5-oxopentyl)amino)ethyl)-5-(3-fluoropropyl)-2,3-dimethoxybenzamide (**5b**) **5b** was synthesized using **4b** (42 mg, 0.1 mmol) in the same procedures as **5a** and obtained 33 mg (88% yield) as a colorless oil. ESI-MS *m/z* calculated for C_21_H_34_FN_2_O_4_^+^ [M+H]^+^ 397.5; found 397.7.

5-Bromo-*N*-(2-(ethyl(4-oxobutyl)amino)ethyl)-2,3-dimethoxybenzamide (**5c**) **5c** was synthesized using **4c** (74 mg, 0.17 mmol) in the same procedures as **5a** and obtained 64 mg (96% yield) as a colorless oil. ESI-MS *m/z* calculated for C_17_H_25_BrN_2_O_4_^+^ [M]^+^ 401.3; found 401.4.

5-Bromo-*N*-(2-(ethyl(5-oxopentyl)amino)ethyl)-2,3-dimethoxybenzamide (**5d**) **5d** was synthesized using **4d** (90 mg, 0.2 mmol) in the same procedures as **5a** and obtained 63 mg (78% yield) as a colorless oil. ESI-MS *m/z* calculated for C_18_H_27_BrN_2_O_4_^+^ [M]^+^ 415.3; found 415.4.

*N*-(2-((4-(Dimethylamino)butyl)(ethyl)amino)ethyl)-5-(3-fluoropropyl)-2,3-dimethoxybenzamide (**6a**) In a mixture of **5a** (74 mg, 0.19 mmol) and 2 M solution of dimethylamine in THF (0.97 mL, 1.93 mmol) in 2 mL of dichloroethane, sodium triacetoxyborohydride (205 mg, 0.97 mmol) was added. The mixture was stirred at RT for 16 h. After completion of the reaction, the mixture was diluted with EtOAc and washed by aq saturated NaHCO_3_ solution and brine. The organic layer was dried over MgSO_4_, filtered and concentrated in vacuo. The crude product was purified by flash chromatography on silica gel (CH_2_Cl_2_/7 N NH_3_ in MeOH = 20:1) to afford 6a (33 mg, 42% yield) as a colorless oil. (^1^H NMR, 400 MHz, MeOD): δ = 7.30 (d, *J* = 1.9 Hz, 1H), 7.06 (d, *J* = 1.9 Hz, 1H), 4.51 (t, *J* = 5.9 Hz, 1H), 4.39 (t, *J* = 5.9 Hz, 1H), 3.90 (s, 3H), 3.88 (s, 3H), 3.51 (d, *J* = 6.6 Hz, 2H), 2.76–2.69 (m, 4H), 2.65 (q, *J* = 7.1 Hz, 2H), 2.56 (t, *J* = 6.9 Hz, 2H), 2.33 (t, *J* = 7.3 Hz, 2H), 2.22 (s, 6H), 2.07–1.94 (m, 2H), 1.54–1.50 (m, 4H), 1.10 (t, *J* = 7.1 Hz, 3H) (^13^C NMR, 100 MHz, MeOD): δ = 168.1, 154.3, 147.3, 139.2, 128.1, 122.3, 117.1, 84.9, 83.2, 62.0, 60.7, 56.7, 54.5, 53.2, 45.5, 38.6, 33.5, 33.3, 32.3_2_, 32.2_7_, 26.3, 26.1, 12.0; ESI-MS *m/z* calculated for C_22_H_39_FN_3_O_3_^+^ [M+H]^+^ 412.6; found 412.6 HRMS (ESI) for C_22_H_39_FN_3_O_3_^+^ [M+H]^+^ requires 412.2975; found 412.2972.

*N*-(2-((5-(Dimethylamino)pentyl)(ethyl)amino)ethyl)-5-(3-fluoropropyl)-2,3-dimethoxybenzamide (**6b**) **6b** was synthesized using **5b** (33 mg, 0.08 mmol) in the same procedures as 6a and obtained 14 mg (42% yield) as a colorless oil. (^1^H NMR, 400 MHz, MeOD): δ = 7.31 (d, *J* = 1.9 Hz, 1H), 7.07 (d, *J* = 1.8 Hz, 1H), 4.51 (t, *J* = 5.9 Hz, 1H), 4.40 (t, *J* = 5.9 Hz, 1H), 3.91 (s, 3H), 3.88 (s, 3H), 3.51 (d, *J* = 6.6 Hz, 2H), 2.77–2.69 (m, 4H), 2.65 (q, *J* = 7.2 Hz, 2H), 2.54 (t, *J* = 7.4 Hz, 2H), 2.31 (t, *J* = 7.7 Hz, 2H), 2.23 (s, 6H), 2.08–1.95 (m, 2H), 1.58–1.48 (m, 4H), 1.38–1.31 (m, 2H), 1.10 (t, *J* = 7.1 Hz, 3H) (^13^C NMR, 100 MHz, MeOD): δ = 168.1, 154.3, 147.3, 139.2, 128.1, 122.4, 117.2, 84.9, 83.2, 62.0, 60.7, 56.7, 54.5, 53.2, 45.4, 38.6, 33.5, 33.3, 32.3_3_, 32.2_7_, 28.2, 28.0, 26.5, 22.2, 12.0; ESI-MS *m/z* calculated for C_23_H_41_FN_3_O_3_^+^ [M+H]^+^ 426.6; found 426.6 HRMS (ESI) for C_23_H_41_FN_3_O_3_^+^ [M+H]^+^ requires 426.3132; found 426.3136.

5-Bromo-*N*-(2-((4-(dimethylamino)butyl)(ethyl)amino)ethyl)-2,3-dimethoxybenzamide (**6c**) **6c** was synthesized using **5c** (64 mg, 0.16 mmol) in the same procedures as **6a** and obtained 34 mg (49% yield) as a colorless oil. (^1^H NMR, 400 MHz, MeOD): δ = 7.53 (d, *J* = 2.3 Hz, 1H), 7.31 (d, *J* = 2.3 Hz, 1H), 3.89 (s, 3H), 3.88 (s, 3H), 3.48 (d, *J* = 6.5 Hz, 2H), 2.68 (d, *J* = 6.5 Hz, 2H), 2.62 (q, *J* = 7.1 Hz, 2H), 2.53 (t, *J* = 6.7 Hz, 2H), 2.32 (t, *J* = 7.2 Hz, 2H), 2.22 (s, 6H), 1.50 (t, *J* = 3.5 Hz, 4H), 1.07 (t, *J* = 7.1 Hz, 3H) (^13^C NMR, 100 MHz, MeOD): δ = 166.5, 155.4, 148.4, 130.0, 125.3, 119.7, 117.7, 62.1, 60.7, 57.1, 54.5, 53.1, 45.4, 38.7, 26.3, 26.1, 12.0; ESI-MS *m/z* calculated for C_19_H_32_BrN_3_O_3_^+^ [M]^+^ 430.4; found 430.5 HRMS (ESI) for C_19_H_32_BrN_3_O_3_^+^ [M]^+^ requires 430.1705; found 430.1703.

5-Bromo-*N*-(2-((5-(dimethylamino)pentyl)(ethyl)amino)ethyl)-2,3-dimethoxybenzamide (**6d**) **6d** was synthesized using **5d** (63 mg, 0.15 mmol) in the same procedures as 6a and obtained 31 mg (46% yield) as a colorless oil. (^1^H NMR, 400 MHz, MeOD): δ = 7.54 (d, *J* = 2.4 Hz, 1H), 7.31 (d, *J* = 2.3 Hz, 1H), 3.89 (s, 3H), 3.88 (s, 3H), 3.48 (d, *J* = 6.5 Hz, 2H), 2.67 (d, *J* = 6.5 Hz, 2H), 2.62 (q, *J* = 7.2 Hz, 2H), 2.51 (t, *J* = 7.4 Hz, 2H), 2.27 (t, *J* = 7.7 Hz, 2H), 2.21 (s, 6H), 1.55–1.45 (m, 4H), 1.35–1.28 (m, 2H), 1.07 (t, *J* = 7.1 Hz, 3H) (^13^C NMR, 100 MHz, MeOD): δ = 166.5, 155.3, 148.4, 130.0, 125.3, 119.7, 117.7, 62.1, 60.8, 57.1, 54.5, 53.0, 45.5, 38.7, 28.3, 28.1, 26.6, 12.0; ESI-MS *m/z* calculated for C_20_H_34_BrN_3_O_3_^+^ [M]^+^ 444.4; found 444.5 HRMS (ESI) for C_20_H_34_BrN_3_O_3_^+^ [M]^+^ requires 444.1862; found 444.1872.

5-Bromo-*N*-(2-((3-(1,3-dioxoisoindolin-2-yl)propyl)(ethyl)amino)ethyl)-2,3-dimethoxybenzamide (**7a**) In a solution of **3b** (2 g, 6.04 mmol) in 20 mL of DMF, *N*-(3-bromopropyl)phthalimide (3.4 g, 12.08 mmol) and K_2_CO_3_ (2.1 g, 15.1 mmol) were added. The mixture was heated to 65 °C and stirred for 16 h. The reaction mixture was cooled to RT and diluted with EtOAc. The mixture was washed by aq saturated NaHCO_3_ solution, water and brine. The organic layer was dried over MgSO_4_, filtered and concentrated in vacuo. The crude product was purified by flash chromatography on silica gel (CH_2_Cl_2_/7 N NH_3_ in MeOH = 40:1) to afford **7a** (1.82 g, 57% yield) as a white solid. (^1^H NMR, 400 MHz, acetone-d_6_): δ = 8.32 (br, 1H), 7.82 (s, 4H), 7.66 (d, *J* = 2.4 Hz, 1H), 7.28 (d, *J* = 2.5 Hz, 1H), 3.92 (s, 3H), 3.91 (s, 3H), 3.73 (t, *J* = 7.1 Hz, 2H), 3.47 (q, *J* = 6.0 Hz, 2H), 2.66 (t, *J* = 6.2 Hz, 2H), 2.62–2.58 (m, 4H), 1.91–1.84 (m, 2H), 1.02 (t, *J* = 7.2 Hz, 3H) (^13^C NMR, 100 MHz, acetone-d_6_): δ = 168.9, 163.7, 154.9, 148.1, 135.0, 133.3, 129.7, 125.5, 123.7, 119.0, 116.9, 61.8, 57.0, 53.2, 51.6, 47.9, 38.4, 36.9, 27.1, 12.0; ESI-MS *m/z* calculated for C_24_H_28_BrN_3_O_5_^+^ [M]^+^ 518.4; found 518.5.

5-Bromo-*N*-(2-((4-(1,3-dioxoisoindolin-2-yl)butyl)(ethyl)amino)ethyl)-2,3-dimethoxybenzamide (**7b**) **7b** was synthesized using **3b** (180 mg, 0.54 mmol) and *N*-(4-bromobutyl)phthalimide (305 mg, 1.08 mmol) in the same procedures as 7a and obtained 220 mg (77% yield) as a colorless oil. (^1^H NMR, 400 MHz, acetone-d_6_): δ = 8.32 (br, 1H), 7.83 (s, 4H), 7.67 (d, *J* = 2.4 Hz, 1H), 7.26 (d, *J* = 2.5 Hz, 1H), 3.90 (s, 3H), 3.87 (s, 3H), 3.73 (t, *J* = 7.1 Hz, 2H), 3.46 (q, *J* = 5.8 Hz, 2H), 2.65 (t, *J* = 6.1 Hz, 2H), 2.62–2.54 (m, 4H), 1.76–1.69 (m, 2H), 1.58–1.51 (m, 2H), 1.04 (t, *J* = 7.1 Hz, 3H) (^13^C NMR, 100 MHz, acetone-d_6_): δ = 168.9, 163.7, 154.9, 148.1, 135.0, 133.2, 129.6, 125.5, 123.7, 119.1, 116.9, 61.8, 57.0, 53.5, 53.1, 48.1, 38.4, 27.2, 25.3, 12.2, 0.1; ESI-MS *m/z* calculated for C_25_H_30_BrN_3_O_5_^+^ [M]^+^ 532.4; found 532.5.

*N*-(2-((3-Aminopropyl)(ethyl)amino)ethyl)-5-bromo-2,3-dimethoxybenzamide (**8a**) In a solution of **7a** (1.82 g, 3.42 mmol) in 34 mL of EtOH, hydrazine hydrate (519 µL, 10.25 mmol) was added. The mixture was heated at 75 °C for 3 h and cooled to RT. The mixture was diluted with EtOAc and washed by aq saturated NaHCO_3_ solution and brine. The organic layer was dried over MgSO_4_, filtered and concentrated in vacuo. The crude product was purified by flash chromatography on silica gel (CH_2_Cl_2_/7 N NH_3_ in MeOH = 10:1) to afford **8a** (1.03 g, 78% yield) as a yellow oil. (^1^H NMR, 400 MHz, MeOD): δ = 7.52 (d, *J* = 2.3 Hz, 1H), 7.31 (d, *J* = 2.4 Hz, 1H), 3.89 (d, *J* = 2.7 Hz, 6H), 3.49 (t, *J* = 6.6 Hz, 2H), 2.68 (t, *J* = 7.0 Hz, 4H), 2.63 (q, *J* = 7.2 Hz, 2H), 2.57 (t, *J* = 7.2 Hz, 2H), 1.70–1.63 (m, 2H), 1.08 (t, *J* = 7.1 Hz, 3H) (^13^C NMR, 100 MHz, MeOD): δ = 166.6, 155.4, 148.3, 130.1, 125.2, 119.8, 117.7, 62.1, 57.1, 53.2, 52.4, 48.6, 41.1, 38.7, 30.9, 12.0; ESI-MS *m/z* calculated for C_16_H_26_BrN_3_O_3_^+^ [M]^+^ 388.3; found 388.4.

*N*-(2-((4-Aminobutyl)(ethyl)amino)ethyl)-5-bromo-2,3-dimethoxybenzamide (**8b**) **8b** was synthesized using **7b** (220 mg, 0.41 mmol) in the same procedures as **8a** and obtained 165 mg (77% yield) as a yellow oil. (^1^H NMR, 400 MHz, MeOD): δ = 7.53 (d, *J* = 2.2 Hz, 1H), 7.30 (d, *J* = 2.2 Hz, 1H), 3.89 (s, 3H), 3.88 (s, 3H), 3.47 (t, *J* = 6.6 Hz, 2H), 2.69–2.59 (m, 6H), 2.52 (t, *J* = 6.6 Hz, 2H), 1.56–1.44 (m, 4H), 1.07 (t, *J* = 7.1 Hz, 3H) (^13^C NMR, 100 MHz, MeOD): δ = 166.5, 155.3, 148.3, 130.0, 125.3, 119.7, 117.7, 62.1, 57.1, 54.5, 53.1, 48.7, 42.6, 38.7, 31.8, 25.5, 12.1; ESI-MS *m/z* calculated for C_17_H_28_BrN_3_O_3_^+^ [M]^+^ 402.3; found 402.4.

5-Bromo-*N*-(2-(ethyl(3-(4-(thiophen-2-yl)benzamido)propyl)amino)ethyl)-2,3-dimethoxybenzamide (**9a**) Thionyl chloride (1.38 mL, 18.9 mmol) was added to 4-(thiophen-2-yl)benzoic acid (129 mg, 0.63 mmol) in a vial. The mixture was stirred at RT for 3 h and the volatiles were removed under the reduced pressure. **8a** (163 mg, 0.42 mmol) in 4.2 mL of CH_2_Cl_2_ and Et_3_N (0.15 mL, 1.05 mmol) were added and the mixture was stirred for 16 h. After completion of the reaction, the volatiles were removed under the reduced pressure and the crude product was purified by flash chromatography on silica gel (CH_2_Cl_2_/7 N NH_3_ in MeOH = 40:1) to afford **9a** (120 mg, 50% yield) as a yellow oil. (^1^H NMR, 400 MHz, MeOD): δ = 7.78 (d, *J* = 8.6 Hz, 2H), 7.67 (d, *J* = 8.5 Hz, 2H), 7.50 (d, *J* = 2.4 Hz, 1H), 7.47 (dd, *J_1_* = 3.6 Hz, *J_2_* = 1.0 Hz, 1H), 7.43 (dd, *J_1_* = 5.1 Hz, *J_2_* = 1.0 Hz, 1H), 7.24 (d, *J* = 2.4 Hz, 1H), 7.11 (dd, *J_1_* = 5.0 Hz, *J_2_* = 3.7 Hz, 1H), 3.85 (s, 3H), 3.83 (s, 3H), 3.50 (t, *J* = 6.4 Hz, 2H), 3.43 (t, *J* = 6.8 Hz, 2H), 2.70 (t, *J* = 6.4 Hz, 2H), 2.67**–**2.62 (m, 4H), 1.86**–**1.79 (m, 2H), 1.07 (t, *J* = 7.1 Hz, 3H) (^13^C NMR, 100 MHz, MeOD): δ = 169.6, 166.7, 155.3, 148.3, 144.3, 138.9, 134.4, 129.9, 129.5, 129.1, 127.2, 126.6, 125.6, 125.3, 119.8, 117.7, 62.1, 57.0, 53.4, 52.3, 39.7, 38.8, 27.8, 11.9; ESI-MS *m/z* calculated for C_27_H_33_BrN_3_O_4_S^+^ [M+H]^+^ 575.5; found 575.5 HRMS (ESI) for C_27_H_33_BrN_3_O_4_S^+^ [M+H]^+^ 574.1375; found 574.1381.

5-Bromo-*N*-(2-(ethyl(4-(4-(thiophen-2-yl)benzamido)butyl)amino)ethyl)-2,3-dimethoxybenzamide (**9b**) **9b** was synthesized using **8b** (120 mg, 0.3 mmol) in the same procedures as **9a** and obtained 35 mg (20% yield) as a colorless oil. (^1^H NMR, 400 MHz, MeOD): δ = 7.80 (d, *J* = 8.5 Hz, 2H), 7.68 (d, *J* = 8.6 Hz, 2H), 7.52 (d, *J* = 2.4 Hz, 1H), 7.49 (dd, *J_1_* = 3.6 Hz, *J_2_* = 1.1 Hz, 1H), 7.45 (dd, *J*_1_ = 5.1 Hz, *J*_2_ = 1.1 Hz, 1H), 7.24 (d, *J* = 2.4 Hz, 1H), 7.12 (dd, *J*_1_ = 5.1 Hz, *J*_2_ = 3.6 Hz, 1H), 3.83 (s, 3H), 3.82 (s, 3H), 3.49 (t, *J* = 6.6 Hz, 2H), 3.41 (t, *J* = 6.6 Hz, 2H), 2.70 (t, *J* = 6.5 Hz, 2H), 2.64 (q, *J* = 7.2 Hz, 2H), 2.59 (t, *J* = 7.4 Hz, 2H), 1.68–1.56 (m, 4H), 1.08 (t, *J* = 7.1 Hz, 3H) (^13^C NMR, 100 MHz, MeOD): δ = 169.7, 166.6, 155.3, 148.3, 144.3, 138.9, 134.5, 130.0, 129.5, 129.2, 127.2, 126.6, 125.6, 125.3, 119.8, 117.7, 62.1, 57.0, 54.2, 53.1, 48.8, 40.8, 38.6, 28.5, 25.5, 11.9; ESI-MS *m/z* calculated for C_28_H_35_BrN_3_O_4_S^+^ [M+H]^+^ 589.6; found 589.4 HRMS (ESI) for C_28_H_35_BrN_3_O_4_S^+^ [M+H]^+^ 588.1532; found 588.1555.

5-Bromo-*N*-(2-(ethyl(4-hydroxybutyl)amino)ethyl)-2,3-dimethoxybenzamide (**10a**) In a solution of **5c** (586 mg, 1.46 mmol) in 15 mL of CH_2_Cl_2_, sodium triacetoxyborohydride (774 mg, 3.65 mmol) was added. The mixture was stirred at RT for 16 h. After completion of the reaction, the mixture was diluted with EtOAc and washed by aq saturated NaHCO_3_ solution and brine. The organic layer was dried over Na_2_SO_4_, filtered and concentrated in vacuo. The crude product was purified by flash chromatography on silica gel (CH_2_Cl_2_/7 N NH_3_ in MeOH = 20:1) to afford 10a (300 mg, 51% yield) as a colorless oil. (^1^H NMR, 400 MHz, MeOD): δ = 7.52 (d, *J* = 2.4 Hz, 1H), 7.30 (d, *J* = 2.4 Hz, 1H), 3.89 (s, 3H), 3.88 (s, 3H), 3.55 (t, *J* = 6.0 Hz, 2H), 3.49 (t, *J* = 6.7 Hz, 2H), 2.70 (t, *J* = 6.7 Hz, 2H), 2.64 (q, *J* = 7.2 Hz, 2H), 2.55 (t, *J* = 7.0 Hz, 2H), 1.61–1.54 (m, 4H), 1.08 (t, *J* = 7.2 Hz, 3H) (^13^C NMR, 100 MHz, MeOD): δ = 166.7, 155.4, 148.3, 130.1, 125.2, 119.8, 117.7, 63.0, 62.1, 57.1, 54.7, 53.1, 48.7, 38.6, 31.9, 24.8, 11.9; ESI-MS *m/z* calculated for C_17_H_28_BrN_2_O_4_^+^ [M+H]^+^ 404.3; found 404.3.

5-Bromo-N-(2-(ethyl(5-hydroxypentyl)amino)ethyl)-2,3-dimethoxybenzamide (**10b**) **10b** was synthesized using **5d** (51 mg, 0.12 mmol) in the same procedures as **10a** and obtained 25 mg (29% yield) as a colorless oil. (^1^H NMR, 400 MHz, MeOD): δ = 7.52 (d, *J* = 2.3 Hz, 1H), 7.30 (d, *J* = 2.4 Hz, 1H), 3.89 (s, 3H), 3.88 (s, 3H), 3.53 (t, *J* = 6.6 Hz, 2H), 3.48 (t, *J* = 6.6 Hz, 2H), 2.69 (t, *J* = 6.6 Hz, 2H), 2.63 (q, *J* = 7.2 Hz, 2H), 2.54 (t, *J* = 7.5 Hz, 2H), 1.58–1.49 (m, 4H), 1.40–1.34 (m, 2H), 1.08 (t, *J* = 7.1 Hz, 3H) (^13^C NMR, 100 MHz, MeOD): δ = 166.7, 155.4, 148.4, 130.1, 125.2, 119.8, 117.7, 63.0, 62.1, 57.1, 54.7, 53.1, 48.8, 38.7, 33.7, 27.9, 25.0, 12.0; ESI-MS *m/z* calculated for C_18_H_29_BrN_2_O_4_^+^ [M]^+^ 417.3; found 417.4.

5-Bromo-*N*-(2-(ethyl(4-((4-methyl-5-phenyl-4*H*-1,2,4-triazol-3-yl)thio)butyl)amino)ethyl)-2,3-dimethoxybenzamide (**11a**) In a mixture of **10a** (100 mg, 0.25 mmol), 4-methyl-5-phenyl-4*H*-1,2,4-triazole-3-thiol (57 mg, 0.3 mmol), and PPh_3_ (98 mg, 0.37 mmol) in 2.5 mL of THF, DIAD (73 µL, 0.37 mmol) was slowly added. The mixture was stirred at RT for 24 h. The mixture was diluted with EtOAc and washed by aq saturated NaHCO_3_ solution and brine. The organic layer was dried over Na_2_SO_4_, filtered and concentrated in vacuo. The crude product was purified by flash chromatography on silica gel (CH_2_Cl_2_/MeOH = 15:1) to afford 11a (23 mg, 16% yield) as a colorless oil. (^1^H NMR, 400 MHz, MeOD): δ = 7.67–7.65 (m, 2H), 7.58–7.54 (m, 3H), 7.51 (d, *J* = 2.4 Hz, 1H), 7.27 (d, *J* = 2.4 Hz, 1H), 4.28 (t, *J* = 6.9 Hz, 2H), 3.85 (s, 6H), 3.61 (s, 3H), 3.50 (t, *J* = 6.4 Hz, 2H), 2.73 (t, *J* = 6.4 Hz, 2H), 2.72–2.63 (m, 4H), 1.98–1.90 (m, 2H), 1.63–1.56 (m, 2H), 1.09 (t, *J* = 7.1 Hz, 3H) (^13^C NMR, 100 MHz, MeOD): δ = 168.5, 166.7, 155.3, 152.4, 148.4, 132.2, 130.3, 129.9, 129.8, 127.3, 125.3, 119.8, 117.7, 62.2, 57.1, 54.0, 53.2, 38.6, 33.7, 27.1, 24.8, 11.8; ESI-MS *m/z* calculated for C_26_H_35_BrN_5_O_3_S^+^ [M+H]^+^ 577.6; found 577.3 HRMS (ESI) for C_26_H_35_BrN_5_O_3_S^+^ [M+H]^+^ 576.1644; found 576.1639.

5-Bromo-*N*-(2-(ethyl(5-((4-methyl-5-phenyl-4*H*-1,2,4-triazol-3-yl)thio)pentyl)amino)ethyl)-2,3-dimethoxybenzamide (**11b**) **11b** was synthesized using **10b** (25 mg, 0.06 mmol) in the same procedures as **11a** and purified by flash chromatography on silica gel (CH_2_Cl_2_/MeOH/7 N NH_3_ in MeOH = 20:1:0.1). 11b was obtained 10 mg (28% yield) as a colorless oil. (^1^H NMR, 400 MHz, MeOD): δ = 7.69–7.67 (m, 2H), 7.58–7.53 (m, 3H), 7.52 (d, *J* = 2.4 Hz, 1H), 7.29 (d, *J* = 2.4 Hz, 1H), 4.23 (t, *J* = 7.0 Hz, 2H), 3.88 (s, 3H), 3.87 (s, 3H), 3.62 (s, 3H), 3.49 (t, *J* = 6.5 Hz, 2H), 2.72 (t, *J* = 6.4 Hz, 2H), 2.66 (q, *J* = 7.1 Hz, 2H), 2.57 (t, *J* = 7.4 Hz, 2H), 1.94–1.87 (m, 2H), 1.63–1.56 (m, 2H), 1.44–1.36 (m, 2H), 1.08 (t, *J* = 7.1 Hz, 3H) (^13^C NMR, 100 MHz, MeOD): δ = 168.4, 166.7, 155.4, 152.4, 148.4, 132.2, 130.3, 130.0, 129.9, 127.4, 125.3, 119.9, 117.7, 62.2, 57.1, 54.4, 53.2, 48.9, 38.6, 33.7, 29.0, 27.3, 25.3, 11.9; ESI-MS *m/z* calculated for C_27_H_36_BrN_5_O_3_S^+^ [M]**^+^** 590.6; found 590.6 HRMS (ESI) for C_27_H_37_BrN_5_O_3_S^+^ [M]+ 590.1800; found 590.1787.

5-Bromo-*N*-(2-(ethyl(methyl)amino)ethyl)-2,3-dimethoxybenzamide (**12**) In a solution of **3b** (40 mg, 0.12 mmol) in 5 mL of acetone, CH_3_I (7.5 µL, 0.12 mmol) and K_2_CO_3_ (36 mg, 0.26 mmol) were added. The mixture was refluxed for 16 h and cooled to RT. The volatiles were removed under the reduced pressure and the crude product was purified by flash chromatography on silica gel (CH_2_Cl_2_/7 N NH_3_ in MeOH = 40:1) to afford 12 (7.8 mg, 19% yield) as a colorless oil. (^1^H NMR, 400 MHz, MeOD): δ = 7.51 (d, *J* = 2.4 Hz, 1H), 7.31 (d, *J* = 2.4 Hz, 1H), 3.89 (s, 3H), 3.88 (s, 3H), 3.52 (t, *J* = 6.7 Hz, 2H), 2.63 (t, *J* = 6.7 Hz, 2H), 2.54 (q, *J* = 7.2 Hz, 2H), 2.31 (s, 3H), 1.11 (t, *J* = 7.2 Hz, 3H) (^13^C NMR, 100 MHz, MeOD): δ = 166.8, 155.4, 148.3, 130.3, 125.2, 119.8, 117.7, 62.1, 57.0, 56.6, 52.6, 41.8, 38.4, 12.4; ESI-MS *m/z* calculated for C_14_H_21_BrN_2_O_3_^+^ [M]^+^ 345.2; found 345.3 HRMS (ESI) for C_14_H_22_BrN_2_O_3_^+^ [M+H]^+^ 345.0814; found 345.0813.

*N*-(3-Bromopropyl)-4-(thiophen-2-yl)benzamide (**13**) **13** was synthesized using 4-(thiophen-2-yl)benzoic acid (150 mg, 0.73 mmol) and 3-bromopropylamine hydrobromide (161 mg, 0.73 mmol) in the same procedures as **9a** and obtained 35 mg (15% yield) as a white solid. (^1^H NMR, 400 MHz, DMSO-d_6_): δ = 8.57 (t, *J* = 5.5 Hz, 1H), 7.89 (d, *J* = 8.5 Hz, 2H), 7.75 (d, *J* = 8.5 Hz, 2H), 7.64–7.61 (m, 2H), 7.17 (dd, *J_1_* = 5.1 Hz, *J_2_* = 3.7 Hz, 1H), 3.59 (t, *J* = 6.6 Hz, 2H), 3.39 (t, *J* = 6.6 Hz, 2H), 2.12–2.05 (m, 2H) (^13^C NMR, 100 MHz, DMSO-d_6_): δ = 166.2, 142.8, 140.2, 136.7, 129.6, 129.2, 128.6, 127.2, 126.0, 125.5, 38.4, 33.0, 18.9; ESI-MS *m/z* calculated for C_14_H_14_BrNOS^+^ [M]^+^ 324.2; found 324.2.

*N*-(3-(Ethyl(methyl)amino)propyl)-4-(thiophen-2-yl)benzamide (**14**) **14** was synthesized using **13** (35 mg, 0.11 mmol) and *N*-methylethanamine (12.8 mg, 0.22 mmol) in the same procedures as **13a** and obtained 18 mg (55% yield) as a white solid. (^1^H NMR, 400 MHz, MeOD): δ = 7.89 (d, *J* = 8.6 Hz, 2H), 7.75 (d, *J* = 8.5 Hz, 2H), 7.52 (dd, *J*_1_ = 3.6 Hz, *J*_2_ = 1.0 Hz, 1H), 7.37 (dd, *J*_1_ = 5.1 Hz, *J*_2_ = 1.0 Hz, 1H), 7.13 (dd, *J*_1_ = 5.1 Hz, *J*_2_ = 3.7 Hz, 1H), 3.58–3.45 (m, 2H), 3.36–3.24 (m, 2H), 3.22–3.11 (m, 2H), 2.88 (s, 3H), 2.10–2.01 (m, 2H), 1.37 (t, *J* = 7.3 Hz, 3H) (^13^C NMR, 100 MHz, MeOD): δ = 170.4, 144.1, 139.4, 133.7, 129.6, 129.3, 127.4, 126.7, 125.8, 54.6, 52.8, 39.8, 37.7, 26.1, 9.7; ESI-MS *m/z* calculated for C_17_H_24_N_2_OS^+^ [M+2H]^+^ 304.5; found 304.2 HRMS (ESI) for C**_17_**H**_23_**N**_2_**OS**^+^** [M+H]**^+^** 303.1531; found 303.1516.

*tert*-Butyl allyl(2-aminoethyl)carbamate (**15a**) In a solution of ethylenediamine (2.8 mL, 41.6 mmol) in 100 mL of CH_2_Cl_2_, ethyl trifluoroacetate (4.9 mL, 41.6 mmol) in 100 mL of CH_2_Cl_2_ was added dropwise at 0 °C. The mixture was warmed to RT and stirred for 1 h. The solvent was removed under the reduced pressure and the residue was dissolved with 210 mL of MeOH. Allyl bromide (3.6 mL, 41.6 mmol) and Et_3_N (6.4 mL, 46 mmol) were added slowly into the mixture and the mixture was stirred for 16 h. Then, (Boc)_2_O (9.6 mL, 41.6 mmol) was added and the mixture was stirred for another 4 h. The volatiles were removed under the reduced pressure and the residue was dissolved EtOAc. The organic layer was washed by aq 0.5 N HCl solution and brine, dried over MgSO_4_, filtered and concentrated in vacuo. Deprotection of trifluoroacetyl group was performed according to the reported method [63] and 2.8 g of **15a** (34% yield) was obtained as a yellow oil. The crude product was used for the next step without further purification.

*tert*-Butyl (2-aminoethyl)(4-fluorobenzyl)carbamate (**15b**) In a solution of *N*-(2-aminoethyl)-2,2,2-trifluoroacetamide (6.5 g, 41.6 mmol) in 200 mL of CH_2_Cl_2_, 4-fluorobenzaldehyde (4.46 mL, 41.6 mmol) and sodium triacetoxyborohydride (17.6 g, 83 mmol) were added. The mixture was stirred for 16 h at RT and washed by aq saturated NaHCO_3_ solution and brine. The organic layer was dried over MgSO_4_, filtered and concentrated in vacuo. The crude product was purified by flash chromatography on silica gel (CH_2_Cl_2_/7 N NH_3_ in MeOH = 20:1) to afford an intermediate (1.7 g, 6.5 mmol) as a colorless oil. Protection of Boc group and deprotection of trifluoroacetyl group were performed according to the reported method [63] and 1.6 g of 15b (15% yield) was obtained as a colorless oil. The crude product was used for the next step without further purification.

*tert*-Butyl allyl(2-(5-bromo-2,3-dimethoxybenzamido)ethyl)carbamate (**16a**) **16a** was synthesized using **1b** (783 mg, 3 mmol) and **15a** (1.2 g, 6 mmol) in the same procedures as **2a** and obtained 783 mg (60% yield) as a colorless oil. (^1^H NMR, 400 MHz, CD_3_CN): δ = 8.04 (br, 1H), 7.69 (d, *J* = 2.3 Hz, 1H), 7.37 (d, *J* = 2.3 Hz, 1H), 5.96–5.87 (m, 1H), 5.26–5.19 (m, 2H), 3.97 (s, 3H), 3.95 (s, 3H), 3.94 (br, 2H), 3.59 (q, *J* = 6.0 Hz, 2H), 3.49 (t, *J* = 5.9 Hz, 2H), 1.48 (s, 9H) (^13^C NMR, 100 MHz, CD_3_CN): δ = 164.8, 154.9, 147.9, 125.1, 119.3, 117.1, 116.7, 80.2, 62.0, 57.3, 39.1, 28.6; ESI-MS *m/z* calculated for C_19_H_27_BrN_2_O_5_^+^ [M]^+^ 443.3; found 443.3.

*tert*-Butyl (2-(5-bromo-2,3-dimethoxybenzamido)ethyl)(4-fluorobenzyl)carbamate (**16b**) **16b** was synthesized using **1b** (500 mg, 1.92 mmol) and **15b** (771 mg, 2.87 mmol) in the same procedures as **2a** and obtained 670 mg (68% yield) as a colorless oil. (^1^H NMR, 400 MHz, CD_3_CN): δ = 7.90 (br, 1H), 7.57 (d, *J* = 2.4 Hz, 1H), 7.29–7.25 (m, 3H), 7.05 (t, *J* = 8.8 Hz, 2H), 4.42 (s, 2H), 3.86 (s, 3H), 3.83 (s, 3H), 3.48 (q, *J* = 6.0 Hz, 2H), 3.39 (br, 2H), 1.38 (s, 9H) (^13^C NMR, 100 MHz, CD_3_CN): δ = 164.9, 164.1, 161.7, 156.8, 155.0, 148.0, 136.1, 130.3, 125.2, 119.4, 117.1, 116.2, 116.0, 80.6, 62.0, 57.3, 46.7, 39.0, 28.6; ESI-MS *m/z* calculated for C_23_H_28_BrFN_2_O_5_^+^ [M]^+^ 511.4; found 511.3.

*N*-(2-(Allylamino)ethyl)-5-bromo-2,3-dimethoxybenzamide (**17a**) **17a** was synthesized using **16a** (350 mg, 0.79 mmol) in the same procedures as **3a** and obtained 250 mg (92% yield) as a colorless oil. (^1^H NMR, 400 MHz, MeOD): δ = 7.45 (d, *J* = 2.4 Hz, 1H), 7.30 (d, *J* = 2.3 Hz, 1H), 5.96–5.86 (m, 1H), 5.26–5.12 (m, 2H), 3.89 (s, 3H), 3.87 (s, 3H), 3.51 (t, *J* = 6.4 Hz, 2H), 3.28 (t, *J* = 1.2 Hz, 1H), 3.27 (t, *J* = 1.3 Hz, 1H), 2.81 (t, *J* = 6.4 Hz, 2H) (^13^C NMR, 100 MHz, CD_3_CN): δ = 167.4, 155.4, 148.1, 137.2, 131.0, 124.9, 119.6, 117.4, 117.1, 62.1, 57.1, 52.9, 40.4; ESI-MS *m/z* calculated for C_14_H_19_BrN_2_O_3_^+^ [M]^+^ 343.2; found 343.5.

5-Bromo-*N*-(2-((4-fluorobenzyl)amino)ethyl)-2,3-dimethoxybenzamide (**17b**) **17b** was synthesized using **16b** (656 mg, 1.28 mmol) in the same procedures as **3a** and obtained 480 mg (91% yield) as a colorless oil. (^1^H NMR, 400 MHz, MeOD): δ = 7.45 (d, *J* = 2.4 Hz, 1H), 7.39–7.35 (m, 2H), 7.29 (d, *J* = 2.4 Hz, 1H), 7.04 (t, *J* = 8.8 Hz, 2H), 3.89 (s, 3H), 3.83 (s, 3H), 3.79 (s, 2H), 3.53 (t, *J* = 6.2 Hz, 2H), 2.83 (t, *J* = 6.2 Hz, 2H) (^13^C NMR, 100 MHz, MeOD): δ = 167.4, 164.9, 162.5, 155.3, 148.1, 136.8, 131.6, 131.5, 130.8, 125.0, 119.6, 117.7, 116.3, 116.1, 62.1, 57.1, 53.5, 40.3; ESI-MS *m/z* calculated for C_18_H_20_BrFN_2_O_3_^+^ [M]^+^ 411.3; found 411.3.

*N*-(2-(Allyl(3-(1,3-dioxoisoindolin-2-yl)propyl)amino)ethyl)-5-bromo-2,3-dimethoxybenzamide (**18a**) **18a** was synthesized using **17a** (240 mg, 0.7 mmol) in the same procedures as **7a** and obtained 100 mg (27% yield) as a colorless oil. (^1^H NMR, 400 MHz, acetone-d_6_): δ = 8.29 (br, 1H), 7.83 (s, 4H), 7.65 (d, *J* = 2.4 Hz, 1H), 7.28 (d, *J* = 2.4 Hz, 1H), 5.92–5.85 (m, 1H), 5.21–5.16 (m, 1H), 5.09–5.06 (m, 1H), 3.92 (s, 3H), 3.91 (s, 3H), 3.72 (t, *J* = 7.1 Hz, 2H), 3.48 (q, *J* = 6.0 Hz, 2H), 3.20 (d, *J* = 1.6 Hz, 2H), 2.68 (t, *J* = 6.2 Hz, 2H), 2.62 (t, *J* = 6.9 Hz, 2H), 1.92–1.85 (m, 2H) (^13^C NMR, 100 MHz, acetone-d_6_): δ = 168.9, 163.8, 154.9, 148.1, 136.7, 135.0, 133.3, 129.8, 125.5, 123.7, 119.0, 117.8, 116.9, 61.8, 57.4, 57.0, 53.4, 51.7, 38.2, 36.8, 26.9; ESI-MS *m/z* calculated for C_25_H_29_BrN_3_O_5_^+^ [M+H]^+^ 531.4; found 531.4.

5-Bromo-*N*-(2-((3-(1,3-dioxoisoindolin-2-yl)propyl)(4-fluorobenzyl)amino)ethyl)-2,3-dimethoxybenzamide (**18b**) **18b** was synthesized using **17b** (270 mg, 0.66 mmol) in the same procedures as **7a** and obtained 333 mg (84% yield) as a colorless oil. (^1^H NMR, 400 MHz, MeOD): δ = 7.80–7.75 (m, 4H), 7.44 (d, *J* = 2.4 Hz, 1H), 7.29 (dd, *J*_1_ = 8.5 Hz, *J*_2_ = 5.5 Hz, 2H), 7.26 (d, *J* = 2.4 Hz, 1H), 6.85 (t, *J* = 8.8 Hz, 2H), 3.87 (s, 3H), 3.84 (s, 3H), 3.68 (t, *J* = 7.0 Hz, 2H), 3.59 (s, 2H), 3.47 (t, *J* = 6.0 Hz, 2H), 2.66 (t, *J* = 6.1 Hz, 2H), 2.53 (t, *J* = 6.8 Hz, 2H), 1.91–1.84 (m, 2H) (^13^C NMR, 100 MHz, MeOD): δ = 170.0, 166.6, 164.6, 162.2, 155.3, 148.3, 135.4, 133.4, 132.0_3_, 130.9_5_, 130.1, 125.3, 124.2, 119.7, 117.7, 116.0, 115.8, 62.2, 58.6, 57.1, 53.9, 52.0, 38.8, 37.1, 27.0; ESI-MS *m/z* calculated for C_29_H_30_BrFN_3_O_5_^+^ [M+H]^+^ 599.5; found 599.3.

*N*-(2-((3-Aminopropyl)(propyl)amino)ethyl)-5-bromo-2,3-dimethoxybenzamide (**19a**) **19a** was synthesized using **18a** (100 mg, 0.19 mmol) in the same procedures as **8a** and obtained 25 mg (33% yield) as a colorless oil. (^1^H NMR, 400 MHz, MeOD): δ = 7.52 (d, *J* = 2.3 Hz, 1H), 7.31 (d, *J* = 2.3 Hz, 1H), 3.89 (s, 3H), 3.88 (s, 3H), 3.48 (t, *J* = 6.5 Hz, 2H), 2.70–2.66 (m, 4H), 2.56 (t, *J* = 7.1 Hz, 2H) 2.50–2.46 (m, 2H), 1.70–1.62 (m, 2H), 1.57–1.47 (m, 2H), 0.91 (t, *J* = 7.4 Hz, 3H) (^13^C NMR, 100 MHz, MeOD): δ = 166.6, 155.3, 148.3, 130.1, 125.2, 119.8, 117.7, 62.1, 57.4, 57.1, 53.9, 53.1, 41.0, 38.8, 30.9, 21.2, 12.3; ESI-MS *m/z* calculated for C_17_H_28_BrN_3_O_3_^+^ [M]^+^ 402.3; found 402.4.

*N*-(2-((3-Aminopropyl)(4-fluorobenzyl)amino)ethyl)-5-bromo-2,3-dimethoxybenzamide (**19b**) **19b** was synthesized using **18b** (156 mg, 0.26 mmol) in the same procedures as 8a and obtained 61 mg (50% yield) as a colorless oil. (^1^H NMR, 400 MHz, MeOD): δ = 7.50 (d, *J* = 2.4 Hz, 1H), 7.34 (dd, *J*_1_ = 8.5 Hz, *J*_2_ = 5.6 Hz, 2H), 7.31 (d, *J* = 2.4 Hz, 1H), 6.97 (7, *J* = 8.8 Hz, 2H), 3.90 (s, 3H), 3.85 (s, 3H), 3.60 (s, 2H), 3.49 (t, *J* = 6.2 Hz, 2H), 2.67–2.61 (m, 4H), 2.53 (t, *J* = 7.0 Hz, 2H), 1.70–1.63 (m, 2H) (^13^C NMR, 100 MHz, MeOD): δ = 166.5, 164.7, 162.3, 155.3, 148.3, 136.6, 132.0, 131.9, 130.0, 125.3, 119.7, 117.7, 116.1, 115.9, 62.2, 58.9, 57.8, 52.5, 40.9, 31.1; ESI-MS *m/z* calculated for C_21_H_27_BrFN_3_O_3_^+^ [M]^+^ 468.4; found 468.5.

5-Bromo-2,3-dimethoxy-*N*-(2-(propyl(3-(4-(thiophen-2-yl)benzamido)propyl)amino)ethyl)benzamide (**20a**) **20a** was synthesized using **19a** (25 mg, 0.06 mmol) in the same procedures as **9a** and obtained 25 mg (71% yield) as a colorless oil. (^1^H NMR, 400 MHz, MeOD): δ = 7.79 (d, *J* = 8.4 Hz, 2H), 7.69 (d, *J* = 8.4 Hz, 2H), 7.50 (d, *J* = 2.4 Hz, 1H), 7.49 (d, *J* = 3.7 Hz, 1H), 7.44 (d, *J* = 5.0 Hz, 1H), 7.25 (d, *J* = 2.4 Hz, 1H), 7.12 (dd, *J*_1_ = 5.0 Hz, *J*_2_ = 3.7 Hz, 1H), 3.86 (s, 3H), 3.84 (s, 3H), 3.50 (t, *J* = 6.3 Hz, 2H), 3.43 (t, *J* = 6.8 Hz, 2H), 2.71 (t, *J* = 6.3 Hz, 2H), 2.64 (t, *J* = 6.8 Hz, 2H), 2.53–2.49 (m, 2H), 1.86–1.79 (m, 2H), 1.57–1.48 (m, 2H), 0.91 (t, *J* = 7.3 Hz, 3H) (^13^C NMR, 100 MHz, MeOD): δ = 169.7, 166.7, 155.3, 148.3, 144.3, 138.9, 134.4, 130.0, 129.5, 129.1, 127.2, 126.6, 125.6, 125.3, 119.8, 117.7, 62.1, 57.4, 57.1, 54.1, 52.9, 39.6, 38.9, 27.9, 21.2, 12.3; ESI-MS *m/z* calculated for C_28_H_35_BrN_3_O_4_S^+^ [M+H]^+^ 589.6; found 589.4 HRMS (ESI) for C_28_H_35_BrN_3_O_4_S^+^ [M+H]^+^ 588.1532; found 588.1527.

*N*-(2-(Allyl(3-(4-(thiophen-2-yl)benzamido)propyl)amino)ethyl)-5-bromo-2,3-dimethoxybenzamide (**20b**) **20b** was synthesized using **13** (36 mg, 0.11 mmol) and **17a** (38 mg, 0.11 mmol) in the same procedures as **7a** and obtained 7 mg (21% yield) as a colorless oil. 20 mg of **17a** (20 mg, 0.06 mmol) was recovered. (^1^H NMR, 400 MHz, MeOD): δ = 7.79 (d, *J* = 8.6 Hz, 2H), 7.69 (d, *J* = 8.5 Hz, 2H), 7.50–7.49 (m, 2H), 7.45 (dd, *J*_1_ = 5.1 Hz, *J*_2_ = 1.0 Hz, 1H), 7.26 (d, *J* = 2.4 Hz, 1H), 7.13 (dd, *J*_1_ = 5.1 Hz, *J*_2_ = 3.7 Hz, 1H), 5.97–5.86 (m, 1H), 5.26–5.21 (m, 1H), 5.18–5.14 (m, 1H), 3.87 (s, 3H), 3.85 (s, 3H), 3.51 (t, *J* = 6.3 Hz, 2H), 3.43 (t, *J* = 6.8 Hz, 2H), 3.22 (d, *J* = 6.6 Hz, 2H), 2.72 (t, *J* = 6.3 Hz, 2H), 2.65 (t, *J* = 6.9 Hz, 2H), 1.87–1.80 (m, 2H) (^13^C NMR, 100 MHz, MeOD): δ = 169.7, 166.7, 155.3, 148.3, 144.3, 138.9, 136.4, 134.4, 130.0, 129.5, 129.1, 127.2, 126.6, 125.6, 125.3, 119.8, 118.8, 117.7, 62.2, 58.2, 57.1, 53.7, 52.5, 39.6, 38.8, 27.9; ESI-MS *m/z* calculated for C_28_H_33_BrN_3_O_4_S^+^ [M+H]^+^ 587.6; found 587.3 HRMS (ESI) for C_28_H_33_BrN_3_O_4_S^+^ [M+H]^+^ 586.1375; found 586.1377.

5-Bromo-*N*-(2-((4-fluorobenzyl)(3-(4-(thiophen-2-yl)benzamido)propyl)amino)ethyl)-2,3-dimethoxybenzamide (**20c**) **20c** was synthesized using **19b** (60 mg, 0.13 mmol) in the same procedures as **9a** and obtained 40 mg (32% yield) as a colorless oil. (^1^H NMR, 400 MHz, MeOD): δ = 7.75 (d, *J* = 8.6 Hz, 2H), 7.68 (d, *J* = 8.6 Hz, 2H), 7.49 (dd, *J*_1_ = 3.6 Hz, *J*_2_ = 1.1 Hz, 1H), 7.47 (d, *J* = 2.4 Hz, 1H), 7.44 (dd, *J*_1_ = 5.1 Hz, *J*_2_ = 1.0 Hz, 1H), 7.33 (dd, *J*_1_ = 8.6 Hz, *J*_2_ = 5.5 Hz, 2H), 7.27 (d, *J* = 2.4 Hz, 1H), 7.12 (dd, *J*_1_ = 5.1 Hz, *J*_2_ = 3.6 Hz, 1H), 6.92 (t, *J* = 8.8 Hz, 2H), 3.86 (s, 3H), 3.83 (s, 3H), 3.62 (s, 2H), 3.50 (t, *J* = 6.0 Hz, 2H), 3.41 (t, *J* = 6.8 Hz, 2H), 2.68 (t, *J* = 6.1 Hz, 2H), 2.60 (t, *J* = 6.8 Hz, 2H), 1.88–1.81 (m, 2H) (^13^C NMR, 100 MHz, MeOD): δ = 169.7, 166.7, 164.7, 162.3, 155.3, 148.3, 144.3, 138.9, 136.5_4_, 136.5_0_, 134.4, 132.1, 132.0, 130.2, 129.6, 129.5, 129.2, 127.2, 126.6, 125.6, 125.2, 119.7, 117.7, 116.1, 115.9, 62.2, 58.8, 57.1, 54.0, 52.3, 39.3, 38.9, 27.8; ESI-MS *m/z* calculated for C_32_H_33_BrFN_3_O_4_S^+^ [M]^+^ 654.6; found 654.6 HRMS (ESI) for C_32_H_34_BrFN_3_O_4_S^+^ [M+H]^+^ 654.1437; found 654.1447.

*N*-(3-((2-(5-Bromo-2,3-dimethoxybenzamido)ethyl)(ethyl)amino)propyl)-2-naphthamide (**21a**) In a solution of **8a** (30 mg, 0.08 mmol) in 2 mL of CH_2_Cl_2_, 2-naphthoyl chloride (22 mg, 0.12 mmol) and Et_3_N (12.9 µL, 0.09 mmol) were added. The reaction mixture was stirred for 1 h at RT followed by 2 mL of MeOH was added. After the mixture was stirred for 10 min, the crude product was purified by flash chromatography on silica gel (CH_2_Cl_2_/7 N NH_3_ in MeOH = 80:1) to afford 21a (37 mg, 86% yield) as a colorless oil. (^1^H NMR, 400 MHz, MeOD): δ = 8.31 (s, 1H), 7.95–89 (m, 3H), 7.83 (dd, *J*_1_ = 8.6 Hz, *J*_2_ = 1.7 Hz, 1H), 7.60–7.53 (m, 2H), 7.48 (d, *J* = 2.4 Hz, 1H), 7.22 (d, *J* = 2.4 Hz, 1H), 3.85 (s, 3H), 3.83 (s, 3H), 3.53–3.48 (m, 4H), 2.72 (t, *J* = 6.4 Hz, 2H), 2.70–2.64 (m, 4H), 1.90–1.83 (m, 2H), 1.09 (t, *J* = 7.1 Hz, 3H) (^13^C NMR, 100 MHz, MeOD): δ = 170.3, 166.7, 155.3, 148.3, 136.4, 134.2, 133.1, 130.1, 130.0, 129.4, 128.9, 128.8, 128.0, 125.2, 124.9, 119.7, 117.7, 62.1, 57.0, 53.4, 52.3, 39.8, 38.8, 27.9, 11.9; ESI-MS *m/z* calculated for C_27_H_32_BrN_3_O_4_^+^ [M]^+^ 542.5; found 542.5 HRMS (ESI) for C_27_H_33_BrN_3_O_4_^+^ [M+H]^+^ 542.1654; found 542.1649.

*N*-(3-((2-(5-Bromo-2,3-dimethoxybenzamido)ethyl)(ethyl)amino)propyl)quinoline-4-carboxamide (**21b**) **21b** was synthesized using **8a** (30 mg, 0.08 mmol) and 4-quinolinecarboxylic acid (21 mg, 0.12 mmol) in the same procedures as **9a** and obtained 27 mg (65% yield) as a colorless oil. (^1^H NMR, 400 MHz, MeOD): δ = 8.90 (d, *J* = 4.4 Hz, 1H), 8.16 (d, *J* = 8.4 Hz, 1H), 8.08 (d, *J* = 8.4 Hz, 1H), 7.84–7.80 (m, 1H), 7.68–7.64 (m, 1H), 7.54 (d, *J* = 4.4 Hz, 1H), 7.50 (d, *J* = 2.4 Hz, 1H), 7.28 (d, *J* = 2.4 Hz, 1H), 3.88 (s, 3H), 3.86 (s, 3H), 3.56–3.49 (m, 4H), 2.75–2.66 (m, 6H), 1.93–1.85 (m, 2H), 1.10 (t, *J* = 7.1 Hz, 3H) (^13^C NMR, 100 MHz, MeOD): δ = 169.8, 166.7, 155.3, 151.2, 149.3, 148.3, 144.6, 131.7, 130.1, 129.9, 129.1, 126.7, 126.1, 125.3, 120.3, 119.8, 117.7, 62.2, 57.1, 53.3, 52.2, 39.5, 38.8, 28.0, 12.0; ESI-MS *m/z* calculated for C_26_H_31_BrN_4_O_4_^+^ [M]^+^ 543.5; found 543.4 HRMS (ESI) for C_26_H_32_BrN_4_O_4_^+^ [M+H]^+^ 543.1607; found 543.1622.

5-Bromo-*N*-(2-(ethyl(3-(4-(pyridin-4-yl)benzamido)propyl)amino)ethyl)-2,3-dimethoxybenzamide (**21c**) **21c** was synthesized using **8a** (30 mg, 0.08 mmol) and 4-(4-pyridyl)benzoic acid (24 mg, 0.12 mmol) in the same procedures as **9a** and obtained 33 mg (75% yield) as a colorless oil. (^1^H NMR, 400 MHz, MeOD): δ = 8.61 (dd, *J*_1_ = 4.6 Hz, *J*_2_ = 1.6 Hz, 2H), 7.92 (d, *J* = 8.4 Hz, 2H), 7.82 (d, *J* = 8.5 Hz, 2H), 7.75 (dd, *J*_1_ = 4.6 Hz, *J*_2_ = 1.6 Hz, 2H), 7.49 (d, *J* = 2.3 Hz, 1H), 7.25 (d, *J* = 2.4 Hz, 1H), 3.87 (s, 3H), 3.84 (s, 3H), 3.51 (t, *J* = 6.4 Hz, 2H), 3.46 (t, *J* = 6.8 Hz, 2H), 2.71 (t, *J* = 6.4 Hz, 2H), 2.70–2.63 (m, 4H), 1.88–1.81 (m, 2H), 1.08 (t, *J* = 7.1 Hz, 3H) (^13^C NMR, 100 MHz, MeOD): δ = 169.5, 166.7, 155.3, 150.9, 149.7, 148.3, 141.8, 136.6, 130.0, 129.3, 128.4, 125.3, 123.4, 119.8, 117.7, 62.1, 57.1, 53.4, 52.3, 40.6, 39.8, 38.8, 27.8, 11.9; ESI-MS *m/z* calculated for C_28_H_33_BrN_4_O_4_^+^ [M]^+^ 569.5; found 569.5 HRMS (ESI) for C_28_H_34_BrN_4_O_4_^+^ [M+H]^+^ 569.1763; found 569.1775.

*N*-(3-((2-(5-Bromo-2,3-dimethoxybenzamido)ethyl)(ethyl)amino)propyl)-1*H*-indole-2-carboxamide (**21d**) **21d** was synthesized using **8a** (30 mg, 0.08 mmol) and indole-2-carboxylic acid (19 mg, 0.12 mmol) in the same procedures as **9a** and obtained 17 mg (42% yield) as a yellow oil. (^1^H NMR, 400 MHz, MeOD): δ = 7.57 (d, *J* = 8.0 Hz, 1H), 7.50 (d, *J* = 2.4 Hz, 1H), 7.42 (dd, *J*_1_ = 8.3 Hz, *J*_2_ = 0.6 Hz, 1H), 7.23 (d, *J* = 2.4 Hz, 1H), 7.22–7.18 (m, 1H), 7.06–7.02 (m, 1H), 7.00 (d, *J* = 0.6 Hz, 1H), 3.86 (s, 3H), 3.83 (s, 3H), 3.51 (t, *J* = 6.4 Hz, 2H), 3.44 (t, *J* = 6.9 Hz, 2H), 2.71 (t, *J* = 6.4 Hz, 2H), 2.68–2.63 (m, 4H), 1.87–1.80 (m, 2H), 1.08 (t, *J* = 7.1 Hz, 3H) (^13^C NMR, 100 MHz, MeOD): δ = 166.8, 164.3, 155.3, 148.3, 138.3, 132.4, 130.0, 129.1, 125.2, 125.1, 122.8, 121.2, 119.7, 117.7, 113.1, 104.2, 62.1, 57.0, 53.4, 52.3, 39.1, 38.8, 28.1, 11.9; ESI-MS *m/z* calculated for C_25_H_31_BrN_4_O_4_^+^ [M]^+^ 531.5; found 531.4 HRMS (ESI) for C_25_H_32_BrN_4_O_4_^+^ [M+H]^+^ 531.1607; found 531.1596.

*N*-(3-((2-(5-Bromo-2,3-dimethoxybenzamido)ethyl)(ethyl)amino)propyl)imidazo[1,2-a]pyridine-2-carboxamide (**21e**) **21e** was synthesized using **8a** (30 mg, 0.08 mmol) and imidazo[1,2-a]pyridine-2-carobxylic acid (20 mg, 0.12 mmol) in the same procedures as 9a and obtained 14 mg (34% yield) as a colorless oil. (^1^H NMR, 400 MHz, MeOD): δ = 8.40 (d, *J* = 6.8 Hz, 1H), 8.13 (s, 1H), 7.45 (d, *J* = 9.2 Hz, 1H), 7.34–7.30 (m, 1H), 7.10 (d, *J* = 2.4 Hz, 1H), 7.04 (d, *J* = 2.3 Hz, 1H), 6.93 (td, *J*_1_ = 6.7 Hz, *J*_2_ = 0.6 Hz, 1H), 3.78 (s, 3H), 3.77 (s, 3H), 3.58 (t, *J* = 5.8 Hz, 2H), 3.50 (t, *J* = 6.3 Hz, 2H), 2.73 (t, *J* = 5.6 Hz, 2H), 2.70–2.66 (m, 4H), 1.86–1.79 (m, 2H), 1.10 (t, *J* = 7.1 Hz, 3H) (^13^C NMR, 100 MHz, MeOD): δ = 167.6, 164.8, 155.0, 147.7, 146.2, 140.4, 131.3, 128.7, 128.1, 124.3, 119.0, 118.5, 117.1, 116.0, 114.9, 62.0, 56.9, 53.9, 53.8, 48.6, 40.4, 38.8, 26.8, 11.8; ESI-MS *m/z* calculated for C_24_H_30_BrN_5_O_4_^+^ [M]^+^ 532.4; found 532.4 HRMS (ESI) for C_24_H_31_BrN_5_O_4_^+^ [M+H]^+^ 532.1559; found 532.1559.

*N*-(3-((2-(5-Bromo-2,3-dimethoxybenzamido)ethyl)(ethyl)amino)propyl)isonicotinamide (**21f**) **21f** was synthesized using **8a** (30 mg, 0.08 mmol) and isonicotinic acid (15 mg, 0.12 mmol) in the same procedures as **9a** and obtained 15 mg (40% yield) as a colorless oil. (^1^H NMR, 400 MHz, MeOD): δ = 8.66 (dd, *J*_1_ = 4.5 Hz, *J*_2_ = 1.7 Hz, 2H), 7.73 (dd, *J*_1_ = 4.5 Hz, *J*_2_ = 1.7 Hz, 2H), 7.49 (d, *J* = 2.4 Hz, 1H), 7.28 (d, *J* = 2.4 Hz, 1H), 3.87 (d, *J* = 2.4 Hz, 6H), 3.50 (t, *J* = 6.5 Hz, 2H), 3.45 (t, *J* = 6.9 Hz, 2H), 2.70 (t, *J* = 6.4 Hz, 2H), 2.67–2.61 (m, 4H), 1.86–1.79 (m, 2H), 1.08 (t, *J* = 7.1 Hz, 3H) (^13^C NMR, 100 MHz, MeOD): δ = 167.7, 166.7, 155.3, 151.1, 148.3, 144.1, 130.1, 125.2, 123.0, 119.8, 117.7, 62.1, 57.1, 53.4, 52.2, 39.7, 38.8, 28.0, 12.0; ESI-MS *m/z* calculated for C_22_H_29_BrN_4_O_4_^+^ [M]^+^ 493.4; found 493.4 HRMS (ESI) for C_22_H_30_BrN_4_O_4_^+^ [M+H]^+^ 493.1450; found 493.1451.

5-Bromo-*N*-(2-(ethyl(3-(4-(thiophen-3-yl)benzamido)propyl)amino)ethyl)-2,3-dimethoxybenzamide (**21g**) **21g** was synthesized using **8a** (30 mg, 0.08 mmol) and 4-(thiophen-3-yl)benzoic acid (25 mg, 0.12 mmol) in the same procedures as 9a and obtained 6 mg (15% yield) as a colorless oil. (^1^H NMR, 400 MHz, MeOD): δ = 7.81 (d, *J* = 8.5 Hz, 2H), 7.74 (d, *J* = 2.1 Hz, 1H), 7.72 (d, *J* = 8.5 Hz, 2H), 7.51 (d, *J* = 2.2 Hz, 2H), 7.50 (d, *J* = 2.4 Hz, 1H), 7.25 (d, *J* = 2.4 Hz, 1H), 3.86 (s, 3H), 3.84 (s, 3H), 3.51 (t, *J* = 6.5 Hz, 2H), 3.45 (t, *J* = 6.8 Hz, 2H), 2.72 (t, *J* = 6.4 Hz, 2H), 2.69–2.63 (m, 4H), 1.87–1.80 (m, 2H), 1.09 (t, *J* = 7.1 Hz, 3H) (^13^C NMR, 100 MHz, MeOD): δ = 169.9, 166.8, 155.3, 148.3, 142.5, 140.3, 134.1, 130.0, 129.0, 127.9, 127.3, 127.2, 125.2, 122.9, 119.8, 117.7, 62.1, 57.1, 54.0, 52.4, 39.7, 38.8, 27.9, 11.9; ESI-MS *m/z* calculated for C_27_H_33_BrN_3_O_4_S^+^ [M+H]^+^ 575.5; found 575.2 HRMS (ESI) for C_27_H_33_BrN_3_O_4_S^+^ [M+H]^+^ 574.1375; found 574.1381.

5-Bromo-*N*-(2-((3-(3-(dimethylamino)benzamido)propyl)(ethyl)amino)ethyl)-2,3-dimethoxybenzamide (**21h**) **21h** was synthesized using **8a** (30 mg, 0.08 mmol) and 3-dimethylaminobenzoic acid (20 mg, 0.12 mmol) in the same procedures as **9a** and obtained 3 mg (7% yield) as a colorless oil. (^1^H NMR, 400 MHz, MeOD): δ = 8.23 (t, *J* = 1.8 Hz, 1H), 7.99 (d, *J* = 7.9 Hz, 1H), 7.85 (dd, *J*_1_ = 7.9 Hz, *J*_2_ = 2.3 Hz, 1H), 7.70 (t, *J* = 8.0 Hz, 1H), 7.52 (d, *J* = 2.4 Hz, 1H), 7.32 (d, *J* = 2.4 Hz, 1H), 3.91 (s, 3H), 3.89 (s, 3H), 3.83 (t, *J* = 5.9 Hz, 2H), 3.55 (t, *J* = 6.4 Hz, 2H), 3.47–3.43 (m, 2H), 3.43–3.37 (m, 4H), 3.34 (s, 6H), 2.17–2.10 (m, 2H), 1.39 (t, *J* = 7.2 Hz, 3H) (^13^C NMR, 100 MHz, MeOD): δ = 168.8, 168.3, 155.3, 148.5, 144.7, 137.7, 132.1, 129.8, 129.5, 125.3, 124.8, 121.0, 120.2, 117.6, 62.2, 57.2, 55.0, 53.4, 52.1, 47.1, 38.0, 36.5, 25.6, 9.2; ESI-MS *m/z* calculated for C_25_H_35_BrN_4_O_4_^+^ [M]^+^ 535.5; found 535.4 HRMS (ESI) for C_25_H_36_BrN_4_O_4_^+^ [M+H]^+^ 535.1920; found 535.1934.

*N*-(3-((2-(5-Bromo-2,3-dimethoxybenzamido)ethyl)(ethyl)amino)propyl)thiophen- e-3-carboxamide (**21i**) **21i** was synthesized using **8a** (30 mg, 0.08 mmol) and 3-thiophenecaroboxylic acid (15 mg, 0.12 mmol) in the same procedures as **9a** and obtained 24 mg (63% yield) as a colorless oil. (^1^H NMR, 400 MHz, MeOD): δ = 7.99 (dd, *J*_1_ = 2.7 Hz, *J*_2_ = 1.6 Hz, 1H), 7.50 (d, *J* = 2.4 Hz, 1H), 7.47–7.43 (m, 2H), 7.28 (d, *J* = 2.4 Hz, 1H), 3.87 (s, 6H), 3.50 (t, *J* = 6.5 Hz, 2H), 3.39 (t, *J* = 7.0 Hz, 2H), 2.71 (t, *J* = 6.4 Hz, 2H), 2.68–2.61 (m, 4H), 1.84–1.77 (m, 2H), 1.07 (t, *J* = 7.1 Hz, 3H) (^13^C NMR, 100 MHz, MeOD): δ = 166.7, 165.7, 155.3, 148.3, 138.7, 130.1, 129.7, 127.6, 127.5, 125.2, 119.7, 117.7, 62.1, 57.1, 53.3, 52.2, 48.7, 39.2, 38.7, 27.9, 11.9; ESI-MS *m/z* calculated for C_21_H_29_BrN_3_O_4_S^+^ [M+H]^+^ 499.4; found 499.2 HRMS (ESI) for C_21_H_29_BrN_3_O_4_S^+^ [M+H]^+^ 498.1062; found 498.1060.

*N*-(3-((2-(5-Bromo-2,3-dimethoxybenzamido)ethyl)(ethyl)amino)propyl)-1-methyl-1*H*-indole-2-carboxamide (**21j**) **21j** was synthesized using **8a** (30 mg, 0.08 mmol) and 1-methylindole-2-carboxylic acid (21 mg, 0.12 mmol) in the same procedures as **9a** and obtained 8 mg (20% yield) as a yellow oil. (^1^H NMR, 400 MHz, MeOD): δ = 7.58 (d, *J* = 8.0 Hz, 1H), 7.50 (d, *J* = 2.3 Hz, 1H), 7.43 (d, *J* = 8.4 Hz, 1H), 7.29 (t, *J* = 7.3 Hz, 1H), 7.25 (d, *J* = 2.4 Hz, 1H), 7.10 (t, *J* = 7.6 Hz, 1H), 6.95 (s, 1H), 4.00 (s, 3H), 3.87 (s, 3H), 3.85 (s, 3H), 3.54 (t, *J* = 6.4 Hz, 2H), 3.44 (t, *J* = 6.8 Hz, 2H), 2.76 (t, *J* = 6.3 Hz, 2H), 2.74–2.67 (m, 4H), 1.89–1.82 (m, 2H), 1.12 (t, *J* = 7.1 Hz, 3H) (^13^C NMR, 100 MHz, MeOD): δ = 166.8, 165.2, 155.3, 148.3, 140.6, 133.5, 130.0, 127.8, 125.2, 125.1, 122.9, 121.5, 119.7, 117.7, 111.2, 105.8, 62.1, 57.0, 54.9, 53.4, 52.3, 39.1, 38.7, 31.9, 27.9, 11.8; ESI-MS *m/z* calculated for C_26_H_34_BrN_4_O_4_^+^ [M+H]^+^ 546.5; found 546.3 HRMS (ESI) for C_26_H_34_BrN_4_O_4_^+^ [M+H]^+^ 545.1763; found 545.1745.

### 4.3. Statistical Analysis

#### 4.3.1. Radioligand Binding Assays

***K***i values for D**_2_** and D**_3_** receptors were measured using [^125^I]IABN in human D**_2_** and D**_3_** receptors expressed in HEK cells, respectively. A filtration binding assay was used to characterize membrane-associated receptor binding properties [49]. The details for the procedures were described in the literature [26].

#### 4.3.2. β-Arrestin Recruitment Assay

CHO-K1 cells which were overexpressed human D**_3_** receptors were cultured in assaycomplete^TM^ cell culture kit 107. Cells were seeded at a density of 25,000 cells per well of 96-well plate, and incubated at 5% CO_2_, 37 °C. Two days later, test compounds were dissolved in DMSO, and diluted with 11-point series in phosphate-buffered saline (PBS). Prepared compounds were added to the cells, and it was incubated for 30 min at 5% CO_2_, 37 °C. Then, cells were treated with 30 nM (EC_80_) of dopamine, and the plate was incubated another 90 min. PathHunter^TM^ detection reagent was added to each well, and then plate was incubated for 80 min at RT in the dark. The chemiluminescent signal was measured by PerkinElmer Enspire plate reader (PerkinElmer, Boston, MA). Data were analyzed by Prism followed by non-linear regression.

#### 4.3.3. Molecular Docking and Molecular Dynamics Simulations (MDS)

The 4 compounds with different ***N***-alkyl groups (**9a**, **20a**, **20b**, and **20c**) and the best candidate **21c** were selected and performed for molecular docking and MDS studies on the D_3_ receptor. The protonated status at physiological pH of each compound was predicted by using Open Babel v3.1.0 [65]. Then, the molecular docking studies and MDS were performed by following the previous protocols [29]. In brief, molecular docking studies were performed via the AutoDock 4.2 [66] plugin on PyMOL (pymol.org). The X-ray structure of the D**_3_** receptor (PDB: 3PBL, resolution: 2.89 Å) was obtained from the RCSB Protein Data Bank (www.rcsb.org (accessed on 19 May 2022)). Heteroatoms were removed from the crystal structure and polar hydrogens were added. Non-polar hydrogens were removed from selected compounds. A grid box with a dimension of 30 × 30 × 28.2 Å^3^ was applied for covering OBS and SBS bindings. The Lamarckian Genetic Algorithm with a maximum of 2,500,000 energy evaluations was used to calculated 100 protein–ligand binding poses for each compound. The D_3_ receptor−ligand complex that reproduced the crystallographic ligand binding pose with good docking score was subjected for the evaluation.

The CHARMM-GUI web-sever [67] was used for MDS preparation. The topology and parameter files of protonated compounds were generated by the Ligand Reader and Modeler module [68,69]. The Bilayer Membrane Builder module [70,71] was used for building the MDS system with FF19SB force field. The protein–ligand complexes generated from docking studies were aligned to the D_3_ receptor structure obtained from the Orientations of Protein in Membranes (OPM) database [72], and the POPC membrane were placed by using the OPM D**_3_** receptor model. The protein, ligand, and membrane complexes were solvated in a TIP3P water box, and then Monte-Carlo sampling was used to add 0.15 M NaCl for charge neutralization. The MDS studies were performed via Amber18 [73] on the high-performance computing (HPC) cluster at Center for Biomedical Image Computing and Analytics at the University of Pennsylvania. The input files of system minimization, 6 steps equilibration including 2 steps NVT ensemble and 4 steps NPT ensemble, and 5 copies of 200 ns production run for MDS were generated from the last step of Membrane Builder [70,71] on the CHARMM-GUI web-sever [67].

The 50 to 200 ns of production simulation with a total of 7500 frames (1500 frames of 5 production simulation copies) for each compound were used for further MDS analysis. The interactions between a ligand and protein in the production simulations were calculated by using the software BINANA v2.1 [74].

## 5. Conclusions

A new scaffold was designed based on metoclopramide and identified having high affinity and subtype selectivity for the D_3_ receptor versus the D_2_ receptor. Initially, **9a** having 4-(thiophen-2-yl)benzamide was recognized as a lead compound showing high binding affinity and subtype selectivity for the D_3_ receptor (*K*i D_2_ = 169 nM and D_3_ = 1 nM). Although different aryl carboxamides exhibited excellent binding affinities preferring D_3_ receptors, **21c** was the most potent (IC_50_ = 1.3 nM) for competing with dopamine in the β-arrestin recruitment assay. Furthermore, the comprehensive screening of **21c** revealed the minimal off-target binding for other CNS targets. Molecular docking or MDS demonstrated that interactions between **21c** and the D_3_ receptor were comparable with fallypride that was known for potent D_2_/D_3_ antagonists. These results suggested that **21c** may have a greater potential for competing with synaptic dopamine for binding to the D_3_ receptor. Overall, this novel scaffold can be developed as high-affinity D_3_ receptor antagonists that bind with low affinity at D_2_ receptors and other CNS receptors.

## Data Availability

Not applicable.

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
