# Peer review of "Design and Synthesis of Conformationally Flexible Scaffold as Bitopic Ligands for Potent D3-Selective Antagonists"

_ijms, 2022, doi:10.3390/ijms24010432_

Round 1

Reviewer 1 Report

Manuscript No.: ijms-2107160

Title: Design and synthesis of conformationally-flexible scaffold based bitopic ligands for potent D3 selective antagonists

Journal: IJMS

In this study, the author has studied “Design and synthesis of conformationally-flexible scaffold based bitopic ligands for potent D3 selective antagonists.” This is an engaging article with a robust approach that purposefully questions our knowledge of the subject. However, the presentation of results is somewhat confusing, and the readability of the discussion could be improved. Addressing both these issues will make this interesting paper more impactful. Sentence-making needs to be improved in this manuscript. The English language used in the manuscript needs major improvements as some punctuation, and grammatical mistakes are present. Experimental designs required more clarity. Moreover, research models are not discussed in an understandable manner. Repetition of data is common in the discussion, which reflects that the author needs a more comprehensive way of thinking. The experimental design and statistical analysis units are not described in sufficient detail and should be improved.

Specific comments:

1.      The Abstract needs to be critically revised, please add some background knowledge and add more results.

2.      Please add more strong keywords.

3.      Page 3, line 20-21: “reduced affinity for the D2 20 receptor (Figure 1).” There is no need of a figure in the introduction section.

4.      Page 5: The whole introduction section looks general. Authors are advised to revise the introduction section carefully and add relevant data to support the problem statement and make a connection between each paragraph. The authors described the figures in the introduction section that does not look appropriate. The authors should make a strong connection between D-receptors and ligand identification. Please add relevant literature about D-receptors and ligand identification. Overall, an introduction needs a major revision.

5.      Page 5: What is the research gap and novelty of the present study?

6.      Page 6: There is always a space between a value and a unit (0 °C, 24 h).

7.      The discussion section needs some modifications. This part only contains information regarding results without any critical view. The discussion also needs professional English editing. Please focus on the main topic during the discussion. The limitations are lacking in the present study. An excellent discussion contains an accurate statement of the results, the relevance, and importance of the results, suitable comparisons to similar studies, alternative explanations of the findings, known limitations, and suggestions for future research. Overall, the discussion section is very poor and needs extensive revision.

8.      Page 22, line 7-8: “The other reagents and solvents were purchased from Sigma-Aldrich (MO, U.S.A.)…” Please add the list of chemicals used in the manuscript with purity and formula.

9.      Page 22, line 3-7: “5-(3-Fluoropropyl)-2,3-dimethoxybenzoic acid (7a) and 5-bromo-2,3-dimethoxybenzoic acid (7b) were prepared by the reported methods[38]. For the OBS binding, tert-butyl (2-aminoethyl) ethylcarbamate was prepared via 3-step synthesis according to the literature[39]. 2-(2-Bromoethyl)-1,3-5 dioxolane and 2-(3-bromopropyl)-1,3-dioxolane were commercially available or prepared according to the described method[40].” This is not an appropriate method to mention this, please briefly explain the methods.

10.  Authors are advised to proofread the manuscript to overcome grammatical mistakes.’

11.  Authors are advised to revise a few headings.

12.  Most of the references are outdated; please revise them and add updated data.

13.  The authors are advised to add a list of abbreviations.

Reviewer 2 Report

Manuscript entitled “Design and synthesis of conformationally-flexible scaffold based bitopic ligands for potent D3 selective antagonists” presents a study on synthesis (the authors have synthesized a large group of new substances) and research on affinity for D3 receptors. The publication is interesting and after minor corrections it can be accepted for publication.

In my opinion, the authors should :

add HR MS analyzes to the supplement,

in Scheme 1, change FPr to (CH2)3F or CH2CH2CH2F, FPr is not clear - it does not show which atom of the fragment is attached to the parent molecule (n-propyl, iso-propyl?)

in general, chemical reaction schemes are more readable if the substituents are represented by formulas and not names, e.g for substances 27 (sch5) it would be better to draw substituents Ar=formula not name, especially that in tables and there are formulas, it will be easier to see the structure-activity relationship

from scheme 1 remove n=2 or 3 because it is next to the numbers of individual substances

clearly write in the conclusions which compounds showed the best activity, the conclusions are very general.

Round 2

Reviewer 1 Report

The authors have carefully addressed the comments, but a few limitations remain. 

1. Please follow the MDPI format and revise the manuscript accordingly.

2. There are reference inconsistencies. Somewhere authors give a space between a value and somewhere without space; somewhere, authors use references in superscript, and somewhere not. An example is given below:

Once this efficacy was confirmed, the assay was conducted in antagonist mode to determine the ability of the antagonist to compete with dopamine at the D3 receptor. The results of the antagonist mode assay are reported as IC50 values[51-53].

3. It is recommended to add a heading of "Statistical Analysis" in the methodology section.
